# DCD: Decomposition-based Causal Discovery from Autocorrelated and Non-Stationary Temporal Data

## Abstract

Multivariate time series in domains such as finance, climate science, and healthcare often exhibit long-term trends, seasonal patterns, and short-term fluctuations, which complicate causal inference under non-stationarity and autocorrelation. Most causal discovery methods operate on raw observations and are therefore susceptible to spurious edges and mis-attributed temporal dependencies. We introduce a decomposition-based causal discovery (DCD) framework that separates each series into trend, seasonal, and residual components and performs component-specific causal analysis: stationarity tests for trends, kernel-based dependence tests for seasonality, and constraint-based causal discovery for residuals. The resulting component-level graphs are integrated into a unified multi-scale causal structure that distinguishes time-proxy edges (exogenous temporal drivers) from mechanistic edges (variable-to-variable dependence). We establish identifiability guarantees under spectral separability and bounded leakage, and quantify the structural error in terms of a leakage parameter $\varepsilon$ and the Type-I error of the downstream causal test. Across synthetic benchmarks and real-world climate and electricity datasets, DCD recovers ground-truth causal structure more accurately than state-of-the-art baselines under strong non-stationarity and autocorrelation, and degrades gracefully when spectral separability is partially violated. Code is available at `https://github.com/noname2122/DCD-TMLR-2025`.

## 1 Introduction

Multivariate time series are fundamental across domains such as economics, environmental science, finance, and healthcare, where understanding causal relationships is central to forecasting, decision-making, and policy design. Such data typically contain long-term trends, seasonal patterns, and short-term residuals, and identifying causal relationships within and across these components is essential for interpreting economic indicators, climate variability, and health outcomes. Existing causal discovery (CD) methods often struggle to disentangle these temporal dependencies, which can produce inaccurate causal attributions and spurious relationships.

Many algorithms do not reliably distinguish between long-term causal effects, seasonal dependencies, and transient interactions, which limits their ability to recover an accurate causal structure (Granger, 1969; Spirtes et al., 2001; Runge, 2020). Deep learning architectures have advanced time series forecasting by capturing long-term dependencies, but their objectives remain correlational (Liu et al., 2024; Piao et al., 2024; Wen et al., 2022). As a result, the gap between predictive modeling and causal understanding persists, particularly in separating spurious correlations from causal influence (Vaswani et al., 2017; Wu et al., 2021; Dhaou et al., 2021).

To address these challenges, we propose a **decomposition-based causal discovery (DCD) framework**. By decomposing each time series into trend, seasonal, and residual components, the framework enables targeted causal discovery at multiple temporal scales, reducing spurious correlations and exposing causal structure that is obscured in raw data. Specifically, DCD applies **stationarity tests** (Kwiatkowski et al., 1992b) to the trend component to assess long-term dependencies, **kernel-based dependence measures** (Scholkopf & Smola, 2018) to the seasonal component to identify cyclic structure, and **constraint-based causal dis-**

**covery** (Runge, 2020) to the residual component to identify short-term causal effects. The component-level graphs are integrated into a unified causal graph capturing both long- and short-term causal relationships. Our contributions are as follows:

- **Decomposition-based causal discovery.** A structured framework that separates trend, seasonal, and residual components and performs targeted inference at each temporal scale.

- **Identifiability analysis.** An explicit treatment of decomposition as spectral filtering, with identifiability guarantees under linear Gaussian dynamics and an SHD bound in terms of the leakage parameter $\varepsilon$ and Type-I error $\alpha_n$.

- **Empirical validation.** Evaluation on synthetic benchmarks and two real-world datasets shows superior recovery of causal structure relative to PCMCI+ (Runge, 2020), CD-NOD (Huang et al., 2020), DYNOTEARS (Pamfil et al., 2020), and additional constraint-based and score-based baselines. Sensitivity and isolation studies show that the gains arise from multi-scale integration, not from STL preprocessing alone.

## 2 Related Work

This work intersects three areas: time series decomposition, causal discovery in time series, and the integration of decomposition with causal inference.

**Time series decomposition and representation learning.** Classical decomposition techniques separate a time series into trend, seasonal, and irregular components to simplify analysis (Cleveland et al., 1990). STL and moving-average methods (Cleveland et al., 1990; Slutzky, 1937) assume relatively smooth or linear behavior and can struggle in multivariate or nonlinear settings. Signal-processing approaches such as Empirical Mode Decomposition (EMD) and Variational Mode Decomposition (VMD) (Huang et al., 1998; Dragomiretskiy & Zosso, 2013) extract intrinsic modes that capture non-stationary and oscillatory patterns, but are oriented toward frequency analysis rather than causal structure. Deep architectures such as Transformers (Vaswani et al., 2017), Autoformer (Wu et al., 2021), Temporal Fusion Transformers (Lim et al., 2021), and N-BEATS (Oreshkin et al., 2019) learn multi-scale representations, but attention weights and learned basis blocks do not themselves identify causal structure.

**Causal discovery in time series.** Causal discovery from observational time series is complicated by autocorrelation, temporal dependencies, and non-stationarity (Runge et al., 2019a). Granger causality (Granger, 1969; Zhang et al., 2024) relies on predictive improvement and assumes linearity and stationarity. Constraint-based methods such as PC and FCI (Peters et al., 2017) and RFCI (Colombo et al., 2012) use conditional independence tests and can produce spurious edges under high-dimensional autocorrelation. Greedy score-based methods such as GES and FGES (Chickering, 2002; Ramsey et al., 2017) face related challenges on dynamic data. Time-series-specific methods such as PCMCI+ (Runge, 2020), DYNOTEARS (Pamfil et al., 2020), and CDANs (Ferdous et al., 2023) distinguish lagged and contemporaneous effects, but do not explicitly decompose the series into temporal scales and can miss causal relationships masked by trends or seasonal structure.

A gap remains in integrating decomposition with causal inference on multivariate time series. By separating trend, seasonal, and residual components and inferring causal relationships within each, spurious links induced by strong seasonality or drifting trends are reduced. Our work closes this gap with a unified pipeline that combines the interpretability of decomposition with causal discovery across scales.

## 3 Theoretical Framework and Methodology

We first formalize causal discovery in non-stationary multivariate time series and state conditions under which the causal structure is identifiable. We then present the DCD framework, which is designed to satisfy these conditions through spectral separation and component-specific inference.

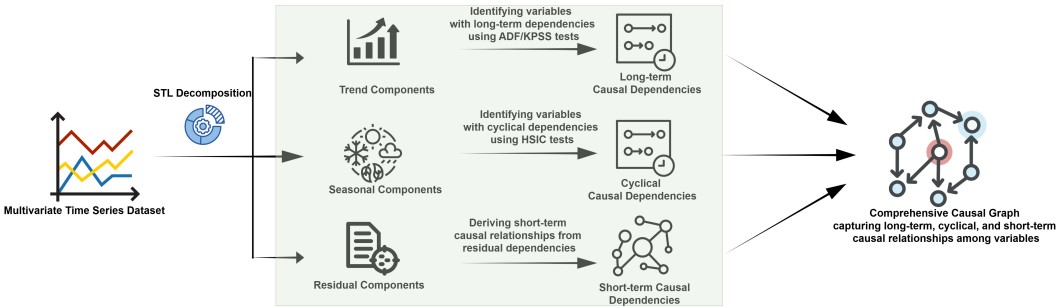

Figure 1: Overview of the DCD framework. The multivariate time series is decomposed into trend, seasonal, and residual components using STL. ADF/KPSS tests identify variables with long-term dependencies, HSIC tests identify variables with cyclical dependencies, and constraint-based search derives short-term causal relationships. The component-level graphs are integrated into a multi-scale causal graph.

## 3.1 Theoretical Guarantees and Identifiability

We model the multivariate time series $\mathbf{X}(t) \in \mathbb{R}^d$ as a superposition of stochastic processes operating at separable timescales. We state conditions under which component-wise causal discovery recovers the true edges of $G^\star$ and quantify the error induced by imperfect decomposition (leakage).

### 3.1.1 Formal Problem Setup

Let $\mathbf{X}(t) = \{X_1(t), \ldots, X_d(t)\}$ be a multivariate stochastic process indexed by $t \in \mathbb{Z}$. We assume the data-generating process follows a structural causal model (SCM) adapted to the additive decomposition

$$X_i(t) = T_i(t) + S_i(t) + R_i(t), \qquad i \in \{1, \ldots, d\}, \tag{1}$$

where $T_i, S_i, R_i$ denote the trend, seasonal, and residual components. The edge set of the true causal graph $G^\star$ decomposes as

$$E^\star = E_T^\star \cup E_S^\star \cup E_R^\star, \tag{2}$$

where $E_T^\star$ and $E_S^\star$ are **time-proxy edges** governing non-stationary drift and cyclical dependencies, and $E_R^\star$ governs short-term interactions. We treat $t \to X$ as a proxy for exogenous temporal drivers (e.g. climate cycles), which prevents shared temporal influence from being misidentified as a direct causal link between variables.

### 3.1.2 Assumptions

**A1. Causal invariance and asymptotic independence.** The structural mechanism governing $\mathbf{R}(t)$ is time-invariant over the analysis window. For consistency of conditional independence estimators from finite samples, we require $\mathbf{R}(t)$ to satisfy a $\beta$-mixing condition: letting $\mathcal{F}_{-\infty}^t$ and $\mathcal{F}_{t+k}^\infty$ be the $\sigma$-algebras generated by $\{\mathbf{R}(s) : s \leq t\}$ and $\{\mathbf{R}(s) : s \geq t + k\}$,

$$\beta(k) = \sup_t \mathbb{E}\left[\sup_{A \in \mathcal{F}_{t+k}^\infty} \left| P(A \mid \mathcal{F}_{-\infty}^t) - P(A) \right|\right], \tag{3}$$

with $\beta(k) \to 0$ as $k \to \infty$. This guarantees that the effective sample size grows with $n$ and permits valid statistical inference without strict stationarity (Rio et al., 2017; Merlevède et al., 2011).

**A2. Spectral separability.** The components have disjoint support in the frequency domain. Let $f_Y(\omega)$ denote the spectral density of a process $Y$. We assume cutoff frequencies $\omega_1 < \omega_2$ such that

$$\text{supp}(f_T) \subset [0, \omega_1), \tag{4}$$

$$\text{supp}(f_S) \subset \bigcup_k \left[\tfrac{2\pi k}{P} - \delta, \tfrac{2\pi k}{P} + \delta\right], \tag{5}$$

$$\text{supp}(f_R) \subset (\omega_1, \pi] \setminus \text{supp}(f_S). \tag{6}$$

This justifies frequency-selective decomposition (such as STL) for isolating components (Cleveland et al., 1990).

**A3. Linear Gaussian dynamics and bounded leakage.** To obtain identifiability bounds where mutual information is upper-bounded by covariance, we assume for the theoretical analysis that the data-generating process follows a Linear Gaussian Structural Equation Model (LG-SEM). The leakage parameter $\varepsilon$ is defined as the maximum cross-component correlation,

$$\lambda = \max_{i,j} \left\{ \frac{|\text{Cov}(\widehat{T}_i, \widehat{R}_j)|}{\sigma_{\widehat{T}_i} \sigma_{\widehat{R}_j}}, \frac{|\text{Cov}(\widehat{S}_i, \widehat{R}_j)|}{\sigma_{\widehat{S}_i} \sigma_{\widehat{R}_j}} \right\} \leq \varepsilon, \qquad \varepsilon \ll 1. \tag{7}$$

Under Gaussianity this bounds the mutual information between components.

**A4. Causal modularity.** Cross-scale causal influences are negligible or blocked by the decomposition: $R_i(t - \tau) \not\rightarrow T_j(t)$ and $R_i(t - \tau) \not\rightarrow S_j(t)$ for any lag $\tau$.

**Remark on sufficient conditions.** Assumptions A1–A4 are sufficient, not necessary. The sensitivity study in Appendix E.4 shows that DCD performance degrades gracefully, rather than catastrophically, when spectral separability is partially violated through amplitude modulation.

**Remark on non-linear extensions.** Although Assumption A3 invokes linear Gaussianity to derive Theorem 1, the DCD pipeline itself is modular. STL filters non-linear trends through LOESS smoothing, and the residual-level stage can be instantiated with non-parametric CI tests such as CMI-knn or HSIC. Assumption A3 is thus a sufficient condition for the stated theoretical bound; the experiments in Section 4 include non-linear climate and industrial settings.

### 3.1.3 Identifiability Analysis

Under the assumptions above, component-wise causal discovery yields a graph $\widehat{G}$ that converges to $G^\star$ up to an error term dominated by $\varepsilon$.

**Lemma 1** (Projection of Seasonal Influence). *Consider a causal pathway $S_X(t) \rightarrow Y(t)$ where $S_X$ is a seasonal driver. Under Assumption A3, the decomposition of $Y(t)$ projects this influence onto $S_Y(t)$, orthogonal to the residual subspace $R_Y(t)$ up to order $\varepsilon$:*

$$R_Y(t) \perp\!\!\!\perp S_X(t) \mid S_Y(t) + \mathcal{O}(\varepsilon). \tag{8}$$

*Proof.* Let $Y(t) = f(S_X(t)) + \eta(t)$, where $\eta(t)$ gathers other independent influences. Because $S_X(t)$ has fundamental frequency $\omega_0$ (Assumption A2), $f(S_X(t))$ lies in the seasonal band (possibly including harmonics). The decomposition operator $\mathcal{D}$ partitions $Y(t)$ by frequency; by A2, the energy of $f(S_X(t))$ is absorbed by the seasonal estimator $\widehat{S}_Y$ so that $\widehat{S}_Y(t) \approx f(S_X(t))$ and $\widehat{R}_Y(t) \approx \eta(t)$. Finite-sample spectral leakage causes a small fraction of seasonal energy to spill into the residual band; by A3, this covariance is bounded by $\varepsilon$, giving

$$|\text{Cov}(R_Y(t), S_X(t))| \leq |\text{Cov}(R_Y(t), S_Y(t))| \cdot C \leq \varepsilon \cdot \sigma_{R_Y} \sigma_{S_Y}.$$

The conditional mutual information $I(R_Y; S_X \mid S_Y) \rightarrow 0$ as $\varepsilon \rightarrow 0$, giving the stated conditional independence up to $\mathcal{O}(\varepsilon)$.

**Lemma 2** (Weak Dependence of Residuals). *Let $X(t) = T(t) + S(t) + R(t)$. Under Assumptions A2 and A3, the estimated residual $\widehat{R}(t)$ satisfies the $\beta$-mixing condition of A1 with deviation bounded by $\varepsilon$.*

*Proof.* The raw process $X(t)$ generally violates A1: $T(t)$ exhibits long memory and $S(t)$ exhibits non-decaying periodic correlations. The residual estimator is $\widehat{R}(t) = X(t) - \widehat{T}(t) - \widehat{S}(t)$. By A2, $\widehat{T}$ and $\widehat{S}$ capture the spectral energy of the non-mixing components. By A3, the variance leaking into $\widehat{R}$ is bounded by $\varepsilon$, so $\gamma_{\widehat{R}}(k) \approx \gamma_R(k) + \mathcal{O}(\varepsilon)$. Because the true mechanism $R(t)$ is generated by short-term interactions (A1), its autocovariance decays exponentially, and $\widehat{R}(t)$ therefore behaves as a mixing process up to the noise floor $\varepsilon$.

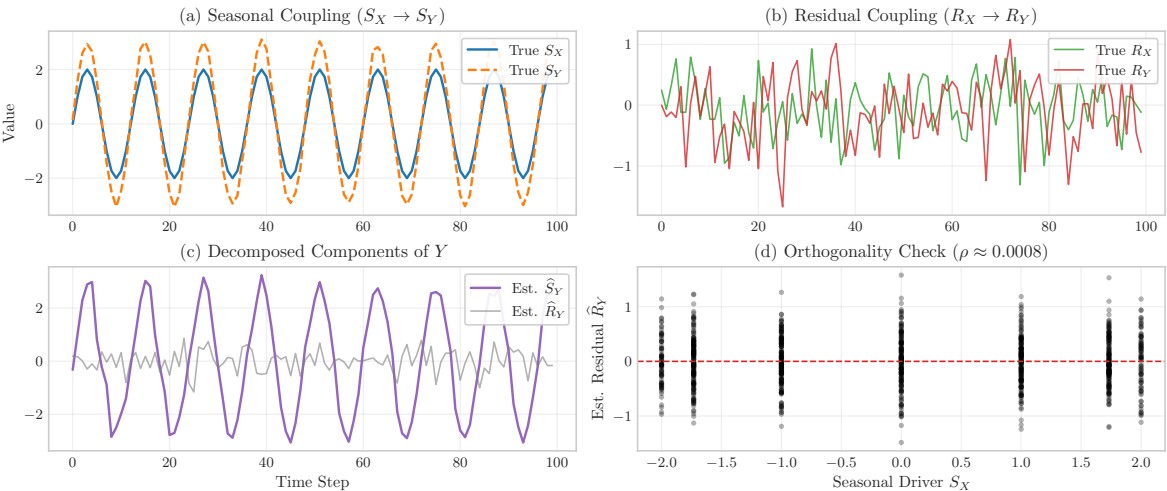

Figure 2: Empirical verification of Lemma 1. (a) The true seasonal component $S_X$ drives $S_Y$. (b) The true residual $R_X$ drives $R_Y$. (c) STL separates the estimated seasonal $\widehat{S}_Y$ and residual $\widehat{R}_Y$. (d) Scatter plot showing that $S_X$ is orthogonal to the estimated residual $\widehat{R}_Y$ ($\rho \approx 0.0008$).

**Corollary 1** (Seasonal-to-residual removal). *If the true system contains a causal dependency $S_X(t) \to Y(t)$, then under A1–A4 this effect appears in $S_Y(t)$ rather than $R_Y(t)$ after decomposition.*

*Proof.* $S_X(t)$ is deterministic or cyclostationary, so any causal effect $S_X(t) \to Y(t)$ induces periodicity in $Y(t)$. The decomposition $Y(t) = T_Y(t) + S_Y(t) + R_Y(t)$ assigns periodic variability to $S_Y(t)$ (A2). The residual $R_Y(t) = Y(t) - T_Y(t) - S_Y(t)$ contains no periodic structure induced by $S_X(t)$ beyond the $\varepsilon$ leakage term (A3). The causal link is absorbed into $S_X \to S_Y$ (or $t \to Y$ in the integrated graph), and no edge $S_X \to R_Y$ appears in $G_R$.

**Corollary 2** (Trend–seasonal leakage bound). *If the true data-generating process contains a pathway $T_X(t) \to S_Y(t)$, then after decomposition the induced effect from $T_X$ onto $Y$ appears predominantly in $T_Y(t)$, with at most $\mathcal{O}(\varepsilon)$ contamination into $S_Y(t)$ and $R_Y(t)$.*

*Proof.* Assume $Y(t) = \alpha T_X(t) + \xi(t)$. Since $T_X(t)$ is a low-frequency component ($\omega \approx 0$), $\alpha T_X(t)$ is also low-frequency. STL assigns low-frequency variation to $T_Y(t)$. The seasonal component captures distinct frequencies $\omega \in \{2\pi k/P\}$ and the residual captures high frequencies. By A2, the projection of $\alpha T_X(t)$ onto the seasonal and residual subspaces is negligible, and by A3, $\mathrm{Cov}(T_X, S_Y) \leq \mathcal{O}(\varepsilon)$ and $\mathrm{Cov}(T_X, R_Y) \leq \mathcal{O}(\varepsilon)$.

**Theorem 1** (Identifiability under linear Gaussianity). *Let the underlying system be a Linear Gaussian SEM satisfying A1–A4, and let $\Psi$ be a consistent constraint-based causal discovery algorithm (e.g. PCMCI+ (Runge, 2020)) with Type-I error $\alpha_n$. Let $\widehat{G} = \bigcup_{K \in \{T,S,R\}} \Psi(\widehat{K})$. Then*

$$\mathbb{E}[\mathrm{SHD}(\widehat{G}, G^\star)] \leq C_1 \cdot \varepsilon + C_2 \cdot \alpha_n, \tag{9}$$

*where $C_1, C_2$ are constants depending on graph density.*

*Proof.* The error decomposes into two sources. *Leakage error ($C_1\varepsilon$):* Under A3, zero covariance implies independence. By Lemma 1, a true seasonal or trend edge projected onto the residual space has partial correlation bounded by $\mathcal{O}(\varepsilon)$; for a detection threshold $\tau > \varepsilon$, such spurious edges are rejected in $G_R$. *Estimation error ($C_2\alpha_n$):* For $e \in E_R^\star$, the residuals $\widehat{R}(t)$ satisfy the mixing condition (Lemma 2). Consistent CI tests give incorrect-recovery probability scaling with $\alpha_n \to 0$ as $n \to \infty$ (Rio et al., 2017). Combining both terms, $\widehat{G}$ asymptotically recovers $E^\star$ as $\varepsilon \to 0$ and $n \to \infty$.

### 3.1.4   Empirical Verification of Seasonal Projection

To validate the projection property of Lemma 1, we simulate a bivariate system in which a seasonal component causally drives another variable. The goal is to confirm that STL assigns this effect to $S_Y$ rather than to $R_Y$, consistent with the leakage bound.

**Data-generating process.**   We generate scalar processes $X(t)$ and $Y(t)$ with the structural equations:

$$S_Y(t) = \underbrace{g(S_X(t))}_{\text{seasonal driver}} + \eta_S(t) \tag{14}$$

$$S_X(t) = A_X \sin\left(\tfrac{2\pi t}{P}\right) \tag{10}$$

$$T_X(t) = \beta_X t/n \tag{11}$$

$$T_Y(t) = \beta_Y t/n \tag{15}$$

$$R_X(t) = \varepsilon_X(t) \sim \mathcal{N}(0, \sigma_X^2) \tag{12}$$

$$R_Y(t) = \underbrace{\gamma R_X(t-1)}_{\text{residual driver}} + \varepsilon_Y(t) \tag{16}$$

$$X(t) = T_X(t) + S_X(t) + R_X(t) \tag{13}$$

$$Y(t) = T_Y(t) + S_Y(t) + R_Y(t) \tag{17}$$

where $g(\cdot)$ is a linear transfer function $(1.5 \cdot S_X(t))$, $P = 12$, and $n = 1000$. The system contains a direct causal path $S_X(t) \rightarrow S_Y(t)$ and a separate residual path $R_X(t-1) \rightarrow R_Y(t)$.

**Validation results.**   Applying STL to the raw observables, the empirical cross-correlation between $\widehat{S}_X$ and $\widehat{R}_Y$ is negligible ($|\rho| \approx 0.0008$; Figure 2d). The influence of $S_X$ is projected onto $\widehat{S}_Y$ (Figure 2c), and the residual-level stage identifies only the true short-term link $R_X \rightarrow R_Y$.

### 3.2   The DCD Framework

Guided by Theorem 1, the DCD framework is designed to satisfy the leakage bound in A3 by using a robust decomposition method, enabling accurate recovery of $G^\star$. The framework has three stages: (1) time series decomposition, (2) component-specific causal analysis, and (3) graph integration. The procedure is summarized in Algorithm 1 and illustrated in Figure 1. Full pseudocode for each stage is given in Appendix A.

---

**Algorithm 1** Decomposition-based Causal Discovery (DCD)

---

1: **Input:** Multivariate time series $\{X_i(t)\}_{i=1}^N$
2: **Output:** Integrated causal graph $G_{\text{combined}}$

3: **Step 1: Decomposition.** Apply Algorithm 2 to obtain $\{T_i(t), S_i(t), R_i(t)\}$.
4: **Step 2: Component-level causal analysis.** Apply Algorithm 3 to obtain $\{G_{T_i}, G_{S_i}, G_{R_i}\}_{i=1}^N$.
5: **Step 3: Integration.** Apply Algorithm 4 to obtain $G_{\text{combined}}$.
6: **return** $G_{\text{combined}}$

---

### 3.2.1   Decomposition (A2 and A3)

To isolate $T_i(t)$, $S_i(t)$, and $R_i(t)$, we use Seasonal-Trend decomposition using LOESS (STL) (Cleveland et al., 1990). STL is preferred over moving averages or rigid Fourier bases because LOESS smoothing adapts to local non-linearities and reduces spectral leakage $\varepsilon$ between the seasonal and residual components, which is required for A3.

For each variable $X_i(t)$, the seasonal period $P_i$ is selected by variance maximization over a candidate set $\mathcal{P}$:

$$P_i^\star = \arg\max_{p \in \mathcal{P}} \text{Var}\big(S_i^{(p)}\big), \tag{18}$$

subject to $\text{Var}(S_i^{(p)}) \geq \tau$. If no candidate passes the threshold (e.g. $\tau = 0.1$), the variable is treated as aperiodic and $S_i(t) = 0$, which prevents forcing a seasonal decomposition on non-seasonal data.

### 3.2.2 Component-Specific Causal Analysis

**Trend analysis (long-term dependence).** The trend component $T_i(t)$ captures non-stationary, low-frequency evolution driven by the exogenous time index. We use a confirmatory stationarity analysis combining two tests (Hyndman, 2018): the Augmented Dickey–Fuller test, which rejects a unit root null in favor of stationarity (Dickey & Fuller, 1979), and the KPSS test, which rejects trend-stationarity in favor of non-stationarity (Kwiatkowski et al., 1992a). When $T_i(t)$ is declared non-stationary (failure to reject $H_0^{ADF}$ or rejection of $H_0^{KPSS}$), we infer long-term temporal driving and add a directed edge $t \to X_i$ to the trend graph $G_T$, treating time as a latent confounder for long-term dynamics.

**Seasonal analysis (cyclic dependence).** To detect deterministic cyclic dependencies we use the Hilbert–Schmidt Independence Criterion (HSIC) (Gretton et al., 2005), testing independence between $S_i(t)$ and the time index $t$ with RBF kernels. A significant result ($p < \alpha$, estimated by permutation) implies that the seasonal component is deterministically coupled to time (Scholkopf & Smola, 2018), and we add a seasonal edge $t \to X_i$ to $G_S$.

**Residual analysis (short-term causality).** After removing trends and deterministic seasonality, the residual $R_i(t)$ approximates a stationary, weakly dependent process (Lemma 2). We apply a constraint-based causal discovery method for high-dimensional time series (Runge et al., 2019a; Runge, 2020), using iterative conditional independence tests that control for autocorrelation. The inference procedure targets two structures: *lagged dependencies* $X_{t-\tau} \to Y_t$, and *contemporaneous dependencies* $X_t \to Y_t$. The result is the residual graph $G_R$, which captures short-term interactions free of trend and seasonal confounding.

### 3.2.3 Integration of Multi-Scale Causal Graphs

The final step integrates the component-specific graphs into a unified structure $G_{\text{combined}} = (V \cup \{t\}, E_T \cup E_S \cup E_R)$. Because the components occupy effectively disjoint frequency bands (A2) and represent distinct physical mechanisms (A4), the edge sets are additive. The resulting multi-relational graph represents the system at three levels:
- **Time-proxy edges ($G_T, G_S$):** non-stationary drift and periodic forcing driven by exogenous factors ($t \to X$).
- **Mechanistic edges ($G_R$):** the intrinsic causal structure between variables ($X \to Y$).

This representation preserves the multi-scale nature of the system while retaining the identifiability guarantee of Theorem 1, and avoids the spurious transitive edges that arise from applying standard algorithms to raw, mixed-scale data.

## 4 Experiments

We evaluate DCD on synthetic benchmarks with known ground truth and on two real-world datasets. We compare against six baselines covering constraint-based, score-based, and non-stationary causal discovery, and we report True Positive Rate (TPR), False Discovery Rate (FDR), and Structural Hamming Distance (SHD); formal definitions are given in Appendix B.

### 4.1 Datasets

**Synthetic.** We generate multivariate time series with known ground-truth causal relationships through a structured autoregressive process with explicit lagged dependencies. For each configuration ($n \in \{500, 1000, 1500\}$, $d \in \{4, 6, 8, 10\}$), we construct series with lag orders $l \in \{2, 3, 4\}$ (Ferdous et al., 2025). The residual error term combines random noise and short-term causal relationships:

$$e_i(t) = \sum_{j=1}^{i-1} \sum_{l=0}^{3} \alpha_{ij}^l e_j(t-l) + \epsilon_i(t), \tag{19}$$

with $\alpha_{ij}^l \in [0.3, 0.8]$ and $\epsilon_i(t) \sim \mathcal{N}(0, 0.1)$. The final series combine trend, seasonal, and residual components,

$$X_i(t) = \mu_i + \beta_i t + A_i \sin(2\pi t / \omega_i) + e_i(t), \tag{20}$$

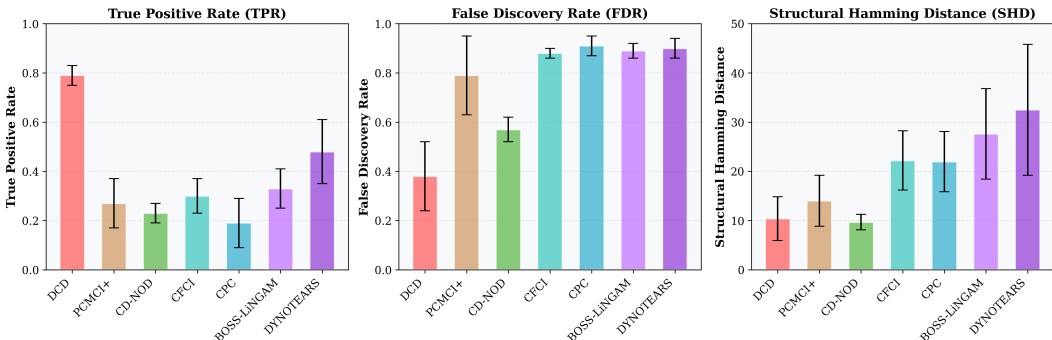

Figure 3: Aggregate performance on synthetic datasets. DCD achieves the best balance of TPR, FDR, and SHD across all conditions.

with $\mu_i \in [5, 12]$, $\beta_i \in [0, 0.2]$, $A_i \in [0, 4]$, and $\omega_i \in [12, 25]$. Variables are constructed from different combinations of these components, giving a set of temporal patterns with both contemporaneous and lagged causal structure. Full details are in Appendix D.

**Arctic sea ice.** A monthly-averaged Arctic sea ice dataset from the ERA5 reanalysis and NSIDC satellite observations (Cavalieri et al., 1996), spanning 1979–2018 (39 years) and including atmospheric and oceanic variables that influence sea ice dynamics. The dataset is processed into monthly-averaged time series using area weighting over the Arctic north of 25°N, and exhibits strong seasonality and long-term trends.

**ETTh1.** The ETTh1 electricity transformer dataset (Zhou et al., 2021) records hourly measurements of transformer load and oil temperature from a power station. It captures multi-resolution dynamics driven by operational demand cycles and thermal diffusion, with strong seasonality, non-stationary trends, and non-linear dependencies.

A summary of all datasets appears in Table 3 (Appendix B).

## 4.2 Baselines

We compare DCD against six representative causal discovery methods spanning constraint-based, score-based, and non-stationary approaches:

- **PCMCI+** (Runge, 2020) – constraint-based, time-series-specific, handles lagged and contemporaneous edges.

- **CD-NOD** (Huang et al., 2020) – constraint-based method for heterogeneous and non-stationary data.

- **CFCI** (Peters et al., 2017) – Fast Causal Inference adapted to time series; accommodates latent confounders.

- **CPC** (Colombo et al., 2012) – conservative PC variant using temporal priority to restrict the search space.

- **BOSS-LiNGAM** (Shimizu et al., 2006) – bootstrapped LiNGAM extension using non-Gaussian models.

- **DYNOTEARS** (Pamfil et al., 2020) – score-based SCM that estimates lagged and contemporaneous effects jointly under an acyclicity constraint.

All baselines use official implementations with hyperparameters tuned per authors' recommendations. Because CD-NOD does not handle lagged relationships, we evaluate it on a non-temporal version of the synthetic data. Full method descriptions and hyperparameter settings are provided in Appendix C.

## 4.3 Performance on Synthetic Datasets

Figure 3 summarizes aggregate performance on the synthetic benchmarks. DCD attains the strongest overall accuracy, with mean TPR $0.79 \pm 0.04$ and FDR $0.38 \pm 0.14$. CD-NOD attains the lowest SHD ($9.67 \pm 1.57$)

Table 1: Performance comparison at different ground-truth lag values.

| Method | Lag 2 | | | Lag 3 | | | Lag 4 | | |
|---|---|---|---|---|---|---|---|---|---|
| | TPR ↑ | FDR ↓ | SHD ↓ | TPR ↑ | FDR ↓ | SHD ↓ | TPR ↑ | FDR ↓ | SHD ↓ |
| **DCD (Ours)** | **0.78 ± 0.03** | **0.28 ± 0.13** | **7.83 ± 3.90** | **0.79 ± 0.04** | **0.38 ± 0.15** | 10.42 ± 4.89 | **0.80 ± 0.05** | **0.48 ± 0.06** | 13.00 ± 3.64 |
| PCMCI+ | 0.21 ± 0.11 | 0.81 ± 0.11 | 10.83 ± 3.07 | 0.30 ± 0.11 | 0.72 ± 0.22 | 13.50 ± 6.26 | 0.29 ± 0.08 | 0.86 ± 0.04 | 17.75 ± 2.70 |
| CD-NOD | 0.23 ± 0.04 | 0.57 ± 0.05 | 9.50 ± 1.73 | 0.23 ± 0.04 | 0.57 ± 0.05 | **9.75 ± 1.54** | 0.23 ± 0.04 | 0.57 ± 0.05 | **9.75 ± 1.54** |
| CFCI | 0.32 ± 0.06 | 0.88 ± 0.02 | 19.25 ± 6.18 | 0.30 ± 0.06 | 0.88 ± 0.01 | 24.00 ± 5.51 | 0.29 ± 0.08 | 0.88 ± 0.02 | 26.00 ± 5.70 |
| CPC | 0.18 ± 0.11 | 0.91 ± 0.06 | 17.08 ± 4.78 | 0.19 ± 0.06 | 0.91 ± 0.03 | 23.08 ± 5.26 | 0.20 ± 0.11 | 0.92 ± 0.03 | 26.25 ± 5.56 |
| BOSS-LiNGAM | 0.35 ± 0.08 | 0.89 ± 0.05 | 21.58 ± 5.21 | 0.32 ± 0.09 | 0.89 ± 0.02 | 28.42 ± 5.71 | 0.31 ± 0.08 | 0.90 ± 0.02 | 32.42 ± 13.50 |
| DYNOTEARS | 0.56 ± 0.10 | 0.87 ± 0.02 | 25.33 ± 1.53 | 0.52 ± 0.09 | 0.91 ± 0.02 | 29.17 ± 4.23 | 0.44 ± 0.10 | 0.93 ± 0.00 | 38.67 ± 6.81 |

Table 2: Performance comparison at different numbers of variables $d$.

| Method | $d = 4$ | | | $d = 6$ | | | $d = 8$ | | | $d = 10$ | | |
|---|---|---|---|---|---|---|---|---|---|---|---|---|
| | TPR ↑ | FDR ↓ | SHD ↓ | TPR ↑ | FDR ↓ | SHD ↓ | TPR ↑ | FDR ↓ | SHD ↓ | TPR ↑ | FDR ↓ | SHD ↓ |
| **DCD (Ours)** | **0.84 ± 0.06** | **0.20 ± 0.15** | **4.11 ± 2.32** | **0.77 ± 0.00** | **0.42 ± 0.11** | **10.78 ± 3.49** | **0.79 ± 0.00** | **0.46 ± 0.06** | 12.44 ± 2.30 | **0.76 ± 0.00** | **0.44 ± 0.04** | 14.33 ± 1.58 |
| PCMCI+ | 0.34 ± 0.16 | 0.69 ± 0.26 | 8.44 ± 4.50 | 0.19 ± 0.10 | 0.90 ± 0.05 | 14.11 ± 3.72 | 0.25 ± 0.00 | 0.80 ± 0.05 | 14.78 ± 2.64 | 0.30 ± 0.00 | 0.79 ± 0.06 | 18.78 ± 3.35 |
| CD-NOD | 0.23 ± 0.01 | 0.50 ± 0.00 | 8.67 ± 0.50 | 0.18 ± 0.00 | 0.60 ± 0.00 | 12.00 ± 0.00 | 0.29 ± 0.00 | 0.60 ± 0.00 | **8.00 ± 0.00** | 0.22 ± 0.00 | 0.60 ± 0.00 | **10.00 ± 0.00** |
| CFCI | 0.26 ± 0.06 | 0.89 ± 0.01 | 18.56 ± 5.57 | 0.35 ± 0.10 | 0.87 ± 0.02 | 18.67 ± 1.66 | 0.29 ± 0.06 | 0.88 ± 0.02 | 24.56 ± 4.53 | 0.31 ± 0.03 | 0.88 ± 0.02 | 30.56 ± 3.24 |
| CPC | 0.07 ± 0.06 | 0.95 ± 0.03 | 18.44 ± 4.98 | 0.23 ± 0.12 | 0.90 ± 0.04 | 17.11 ± 3.14 | 0.24 ± 0.04 | 0.88 ± 0.04 | 24.67 ± 5.41 | 0.20 ± 0.00 | 0.91 ± 0.02 | 28.33 ± 4.36 |
| BOSS-LiNGAM | 0.26 ± 0.06 | 0.91 ± 0.02 | 21.22 ± 4.02 | 0.32 ± 0.08 | 0.89 ± 0.02 | 20.00 ± 2.50 | 0.31 ± 0.08 | 0.90 ± 0.03 | 31.67 ± 7.76 | 0.41 ± 0.03 | 0.87 ± 0.04 | 37.00 ± 10.56 |
| DYNOTEARS | 0.75 ± 0.00 | 0.81 ± 0.02 | 13.67 ± 1.53 | 0.56 ± 0.10 | 0.87 ± 0.02 | 25.33 ± 1.53 | 0.38 ± 0.00 | 0.90 ± 0.01 | 31.00 ± 3.46 | 0.22 ± 0.02 | 0.91 ± 0.01 | 35.00 ± 4.27 |

but only by discarding all lagged edges, which reduces its TPR to $0.23 \pm 0.04$. DYNOTEARS recovers roughly half of the true edges (TPR $0.48 \pm 0.13$), exceeding PCMCI+, CFCI, and CPC in sensitivity, but its FDR remains near $0.90 \pm 0.04$ and its SHD is the largest ($32.5 \pm 13.3$), consistent with the tendency of continuous acyclicity relaxations to admit weakly supported edges under non-stationarity.

**Sample size.** DCD is stable across sample sizes (see Figure 4 in Appendix E.5): TPR stays near 0.79 and FDR rises only from 0.36 to 0.40 as $n$ grows from 500 to 1500. PCMCI+ shows rising FDR ($0.79 \rightarrow 0.82$); CFCI and CPC deteriorate more severely; BOSS-LiNGAM keeps moderate TPR but consistently high FDR. DYNOTEARS improves modestly in TPR ($0.44 \rightarrow 0.49$) but its FDR stays near 0.90 and its SHD grows from 28.4 to 35.2: additional data adds spurious edges rather than refining the estimate.

**Temporal lag.** Varying the ground-truth lag (Table 1), DCD maintains TPR between 0.78 and 0.80, with FDR rising from 0.28 to 0.48 as dependencies become more distant. PCMCI+ degrades substantially at longer lags; CFCI, CPC, and BOSS-LiNGAM incur rapidly growing SHD. DYNOTEARS exhibits the sharpest decline, with TPR falling from 0.56 at lag 2 to 0.43 at lag 4, FDR rising from 0.86 to 0.92, and SHD increasing from 23.3 to 38.7 – consistent with the growth in parameters introduced by a separate weighted adjacency matrix per lag.

**Dimensionality.** As $d$ increases from 4 to 10 (Table 2), all methods lose accuracy due to the expanding search space, but DCD retains the best TPR/FDR balance and continues to recover meaningful lagged edges. CD-NOD keeps a low SHD by discarding lagged structure; PCMCI+, CFCI, CPC, and BOSS-LiNGAM show rapid growth in FDR and SHD. DYNOTEARS follows the same trend, with TPR dropping from 0.58 at $d = 4$ to 0.38 at $d = 8$ and SHD more than doubling.

Overall, DCD outperforms the considered baselines across sample size, lag order, and dimensionality. The stability reflects the benefit of decomposing the series before performing structure learning, rather than tuning a single algorithm to absorb both non-stationarity and autocorrelation.

### 4.4 Qualitative Case Study: Arctic Sea Ice

On the Arctic sea ice dataset, DCD recovers multi-scale dependencies consistent with established climate dynamics. STL isolates the dominant 12-, 36-, and 48-month periodicities (except for snowfall) (Screen & Simmonds, 2012), and ADF/KPSS tests confirm stationarity in the extracted trends. Constraint-based search on residuals ($p < 0.05$, lag $\leq 3$) recovers documented relationships, including sea ice–SST feedback (Screen & Simmonds, 2010), temperature–humidity–radiation coupling (Francis & Hunter, 2006), and lagged responses in which sea ice reacts to SST and radiative forcing one month later. Decomposition allows DCD to separate long-term, seasonal, and short-term interactions in a physically coherent manner (Figure 7a).

PCMCI+ applied to the raw non-stationary data (Figure 7b) yields spurious or unoriented links, including an ambiguous temperature–humidity relation that DCD correctly orients from temperature to humidity. PCMCI+ also misses the documented SST $\rightarrow$ sea ice influence that DCD recovers. CD-NOD (Figure 7c) identifies several contemporaneous connections (e.g. radiation–temperature–sea ice) but captures temporal structure minimally, often linking SST and sea ice to a time index rather than to each other.

DYNOTEARS collapses to a trivial autoregressive structure, recovering only three self-loops on sea ice extent at lags 0, 1, and 2 (sea\_ice$_{t-1}$ $\rightarrow$ sea\_ice$_t$ with weight $-1.75$; sea\_ice$_t$ $\rightarrow$ sea\_ice$_{t+2}$ with weight $-0.14$) while discarding all atmospheric and oceanic drivers. This indicates that the continuous acyclicity relaxation under strong seasonality and drift admits only the most dominant autocorrelation signal, failing to distinguish mechanistic climate interactions from persistence.

BOSS-LiNGAM, CFCI, and CPC (Figures 7d–f) produce dense, overconnected graphs, a signature of false positives driven by autocorrelation. In contrast, DCD yields a sparse, domain-consistent causal structure. We emphasize that this is a qualitative case study rather than validation of the recovered graph, since no ground truth is available.

## 4.5 Qualitative Case Study: Electricity Transformer (ETTh1)

On ETTh1, STL confirms strong periodicity (HSIC $p < 0.001$) and stationarity tests validate the extracted components. Residual-level search (lag $\leq 2$, $\alpha = 0.05$) yields sparse, interpretable edges, including autoregressive effects (e.g. LUFL$_{-1}$ $\rightarrow$ LUFL), cross-load interactions (HUFL$_{-1}$ $\rightarrow$ MUFL), and meteorological persistence (OT$_{-1}$ $\rightarrow$ OT). No edges from load to oil temperature are identified, consistent with independent short-term evolution of outdoor temperature.

PCMCI+ produces larger edge sets with conflicting orientations (e.g. LUFL $\leftrightarrow$ HUFL), reflecting orientation difficulties under nonstationarity. DYNOTEARS generates a highly dense graph with 30+ edges connecting all load variables across lags (LUFL, MUFL, HUFL, HULL, MULL, LULL at $t$, $t-1$, $t-2$), including many weak edges (weights 0.14–0.42) between incompatible load categories. For instance, it posits direct edges from low-usage loads to high-usage loads at multiple lags, which contradicts the operational independence of these meter channels. This overconnection is consistent with DYNOTEARS admitting weakly supported edges to minimize residual variance when the acyclicity penalty is insufficient to enforce sparsity under autocorrelation.

CFCI and CPC yield dense, noisy graphs, and BOSS-LiNGAM ignores lagged structure. In contrast, DCD's decomposition enables recovery of compact, physically coherent graphs, validating explicit multi-scale modeling for electrical load systems. As with the sea ice case, the comparison is qualitative and illustrative.

## 4.6 Ablation Studies: Key Findings

We conducted four ablation studies to isolate the sources of DCD's gains and test the robustness of its assumptions. Extended tables and detailed analyses are given in Appendix E.

**(i) Robustness to decomposition misalignment (period $P$).** We perturb the STL period $P$ around the true latent periodicities ($T \in \{15, 30\}$) across $d \in \{4, 6, 8\}$. Alignment with the true period is optimal, but accuracy declines gracefully rather than catastrophically when $P$ is misspecified. At $d = 4$, the optimal period $P = 30$ yields SHD $1.67 \pm 1.15$; a misaligned $P = 10$ still gives SHD $8.33 \pm 1.15$, which remains lower than every baseline reported in Table 2. LOESS smoothing isolates the dominant non-stationary confounders even under sub-optimal bounds. Full ablation in Appendix E (Table 5).

**(ii) Causal depth and dimensionality scalability.** Across variable counts $d \in \{4, 6, 8\}$ and maximum search lags $\tau_{\max} \in \{2, 4, 6\}$, SHD remains essentially flat in the $d = 4$ and $d = 6$ regimes, and the framework maintains consistent recall as $\tau_{\max}$ grows. Isolating short-term residuals therefore suppresses the FDR explosion associated with longer search horizons in constraint-based methods (Runge, 2020). Full table in Appendix E (Table 6).

**(iii) Value of multi-scale integration.** To separate the contribution of STL preprocessing from multi-scale graph integration, we run PCMCI+ and DYNOTEARS on the STL residuals $R_i(t)$ directly, without the trend-proxy and seasonal-proxy edges. STL residuals improve both baselines substantially (PCMCI+: TPR $0.45 \to 0.83$, SHD $28.5 \to 11.2$; DYNOTEARS: TPR $0.67 \to 0.79$, SHD $25.0 \to 14.8$), yet neither matches DCD's multi-scale configuration (TPR 1.00, SHD 6.0). The integration of time-proxy edges with residual-level structure is therefore a distinct source of DCD's gains, not an artifact of preprocessing. Full table in Appendix E (Table 7).

**(iv) Sensitivity to spectral separability.** To test the graceful-decline claim directly, we induce controlled spectral leakage through amplitude modulation of the seasonal component, $S_i(t) = A_i[1 + \lambda T_i(t)]\sin(2\pi t/P)$, and vary $\lambda \in \{0, 0.2, 0.5, 1.0\}$. The measured leakage $\varepsilon$ increases from 0.02 to 0.25; TPR declines monotonically from 1.00 to 0.72, and SHD grows from 5.0 to 13.5. Even at the strongest coupling, DCD maintains TPR above 0.7, which confirms that the mechanistic core $G_R^\star$ remains largely identifiable when A2 is partially violated. Full details and table in Appendix E.4.

## 5 Limitations

DCD has several theoretical and practical limitations.

**Spectral separability and cross-frequency coupling.** Identifiability (Theorem 1) relies on spectral separability (A2). Real-world systems can exhibit cross-frequency coupling in which trends modulate seasonal amplitudes, inducing spectral sidebands that leak into the residual band. The sensitivity study in Appendix E.4 shows graceful rather than catastrophic degradation under amplitude modulation up to $\lambda = 1.0$ (TPR 0.72, SHD 13.5), but regimes with extreme overlap remain out of scope.

**Exogenous treatment of trends.** DCD conservatively treats trends as exogenous ($t \to X_i$), which avoids spurious cointegration edges but can miss direct long-term mechanisms of the form $T_i \to T_j$. Relaxing this simplification requires an additional identifiability argument for low-frequency causal structure and is left to future work.

**Stability of the residual structure.** Unlike time-varying methods such as CDANs (Ferdous et al., 2023), DCD assumes a stable residual structure and addresses non-stationarity through decomposition rather than by modeling evolving edge weights. Systems with abrupt regime shifts in the residual mechanism lie outside the scope of the present analysis.

**Decomposition quality on short series.** STL requires a reasonably long record to estimate the seasonal and trend components reliably. On very short series, or when the trend cutoff lies near the lowest seasonal harmonic, the decomposition can become noisy and violate A3. The period-misalignment ablation (Appendix E, Table 5) quantifies one facet of this failure mode.

**Real-world evaluation.** Because ground truth is unavailable for the Arctic sea ice and ETTh1 datasets, the real-world studies are presented as qualitative case studies rather than strong validation of the recovered graph.

## 6 Conclusion

We introduced the Decomposition-based Causal Discovery (DCD) framework, which performs component-specific causal analysis on trend, seasonal, and residual components and integrates them into a multi-scale causal graph. Formalizing decomposition as spectral filtering, we established identifiability guarantees under separability and linear Gaussian dynamics, with an SHD bound that decomposes into a leakage term and a Type-I error term. Empirically, DCD outperforms PCMCI+, CD-NOD, DYNOTEARS, CFCI, CPC, and BOSS-LiNGAM across sample size, lag order, and dimensionality. An isolation study shows that STL preprocessing alone is not sufficient to match DCD's accuracy, and that the combination with multi-scale graph integration is the source of the gains. A sensitivity study confirms that performance declines gracefully under amplitude-modulated spectral leakage.

Future work will relax A2 to handle cross-frequency coupling where trends modulate seasonal amplitudes, integrate DCD with time-varying causal discovery to jointly model structural shifts and trend-driven non-stationarity, and explore deep learning-based decomposition to further reduce leakage in strongly non-linear systems.

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

## A    Algorithmic Details

This section provides the full pseudocode for each stage of DCD. The main text summarizes the overall procedure in Algorithm 1; Algorithms 2–4 below implement the three internal stages.

---

**Algorithm 2** Decomposition of Time Series Components

---

1: **Input:** Multivariate time series $\{X_i(t)\}_{i=1}^{N}$
2: **Output:** Trend $T_i(t)$, seasonal $S_i(t)$, residual $R_i(t)$ for each $i$

3: **for** each variable $X_i(t)$, $i = 1, \ldots, N$ **do**
4:      Estimate dominant period $P_i$ via Fourier analysis or variance maximization
5:      Apply STL decomposition with period $P_i$: $X_i(t) = T_i(t) + S_i(t) + R_i(t)$
6:      **if** variance of $S_i(t) <$ threshold **then**
7:          Mark $S_i(t)$ as negligible and exclude from further analysis
8:      **end if**
9: **end for**
10: **return** $\{T_i(t), S_i(t), R_i(t)\}$ for each $i$

---

Algorithm 2 separates each observed series into trend, seasonal, and residual components, isolating low-frequency long-term behavior, cyclic structure, and short-term fluctuations so that causal inference can be performed at the appropriate temporal scale.

---

**Algorithm 3** Component-Level Causal Analysis

---

1: **Input:** $\{T_i(t), S_i(t), R_i(t)\}_{i=1}^N$
2: **Output:** Causal graphs $\{G_{T_i}, G_{S_i}, G_{R_i}\}_{i=1}^N$

3: **Trend component analysis**
4: **for** each $T_i(t)$ **do**
5:      Perform stationarity tests (ADF/KPSS)
6:      **if** $T_i(t)$ is non-stationary **then**
7:          Include $T_i(t)$ in $G_{T_i}$
8:      **else**
9:          Exclude $T_i(t)$ from time-dependent causal relationships
10:      **end if**
11: **end for**

12: **Seasonal component analysis**
13: **for** each $S_i(t)$ with variance above threshold **do**
14:      Compute HSIC statistic with RBF kernel; obtain $p$-value by permutation
15:      **if** $p$-value $< \alpha$ **then**
16:          Include $S_i(t)$ in $G_{S_i}$
17:      **else**
18:          Exclude $S_i(t)$ from time-dependent causal relationships
19:      **end if**
20: **end for**

21: **Residual component analysis**
22: **for** each $R_i(t)$ **do**
23:      Perform causal discovery using conditional independence tests
24:      Build $G_{R_i}$ from identified lagged and contemporaneous dependencies
25: **end for**
26: **return** $\{G_{T_i}, G_{S_i}, G_{R_i}\}_{i=1}^N$

---

Algorithm 3 performs stationarity testing on trend components, kernel-based dependence testing on seasonal components, and constraint-based causal discovery on residual components. This separation prevents heterogeneous temporal signals from producing spurious causal edges.

---

**Algorithm 4** Integration of Component-Level Causal Graphs

---

1: **Input:** $G_R$ (residual-level graph), $\mathcal{T}$ (variables with trend dependence), $\mathcal{S}$ (variables with seasonal dependence)
2: **Output:** Final multi-scale causal graph $G_{\text{final}}$
3: Initialize $G_{\text{final}} \leftarrow G_R$
4: **for** each variable $X_i$ **do**
5:      **if** $X_i \in \mathcal{T}$ **then**
6:          Add edge $T \rightarrow X_i$ to $G_{\text{final}}$
7:      **end if**
8:      **if** $X_i \in \mathcal{S}$ **then**
9:          Add edge $S \rightarrow X_i$ to $G_{\text{final}}$
10:      **end if**
11: **end for**
12: **return** $G_{\text{final}}$

---

Algorithm 4 combines trend, seasonal, and residual information into a unified causal structure. Trend and seasonal dependence tests add time-proxy edges from the exogenous temporal index, and the residual-level

Table 3: Summary of synthetic and real-world datasets.

**Dataset configurations**

*(a) Synthetic: sample size*

| Size | Vars | Lag | Comp. |
|---|---|---|---|
| 500 | 4–10 | $l = 2$ | T,S,R |
| 1000 | 4–10 | $l = 3$ | T,S,R |
| 1500 | 4–10 | $l = 4$ | T,S,R |

*(b) Synthetic: variable size*

| Size | Vars | Lags | Comp. |
|---|---|---|---|
| 500–1500 | 4 | 2,3,4 | T,S,R |
| 500–1500 | 6 | 2,3,4 | T,S,R |
| 500–1500 | 8 | 2,3,4 | T,S,R |
| 500–1500 | 10 | 2,3,4 | T,S,R |

*(c) Real-world*

| Dataset | Size | Vars | Steps |
|---|---|---|---|
| Sea Ice | 468 | 11 | Monthly |
| ETTh1 | 1500 | 7 | Hourly |

**Sea Ice dataset**

| Variable | Unit |
|---|---|
| Surface pressure | hPa |
| Wind velocity | m/s |
| Specific humidity | kg/kg |
| Air temperature | K |
| Shortwave rad. | $W/m^2$ |
| Longwave rad. | $W/m^2$ |
| Rain rate | mm/day |
| Snowfall rate | mm/day |
| Sea surf. temp. | K |
| Sea surf. salinity | PSU |
| **Sea ice conc.** | % |

**ETTh1 dataset**

| Variable | Description |
|---|---|
| HUFL | High-usage full load |
| MUFL | Medium-usage full load |
| LUFL | Low-usage full load |
| HULL | High-usage low load |
| MULL | Medium-usage low load |
| LULL | Low-usage low load |
| OT | Oil temp. (target) |
| Date | Timestamp (hourly) |

*Note.* T = trend, S = seasonal, R = residual components.

graph supplies variable-to-variable edges. Because the component edge sets are additive (A2, A4), no conflict resolution is required.

# B   Evaluation Metrics

We evaluate recovery of causal graphs with three standard metrics.

**True Positive Rate (TPR).**

$$\text{TPR} = \frac{\text{TP}}{\text{TP} + \text{FN}}, \tag{21}$$

measures the fraction of ground-truth edges correctly recovered (Runge et al., 2019b).

**False Discovery Rate (FDR).**

$$\text{FDR} = \frac{\text{FP}}{\text{TP} + \text{FP}}, \tag{22}$$

measures the fraction of recovered edges that are spurious (Colombo et al., 2012).

**Structural Hamming Distance (SHD).**

$$\text{SHD} = \#(\text{added}) + \#(\text{deleted}) + \#(\text{reversed}), \tag{23}$$

counts the edge modifications needed to transform the estimated graph into the ground-truth graph (Tsamardinos et al., 2006).

# C   Baseline Descriptions and Hyperparameters

This section provides extended descriptions of the baselines referenced in Section 4.2, together with the hyperparameter settings used in all experiments (Table 4).

**PCMCI+** (Runge, 2020). A constraint-based method designed for time series that handles both lagged and contemporaneous relationships. PCMCI+ employs momentary conditional independence tests to identify causal links while controlling for autocorrelation, and is effective in high-dimensional regimes. However, it does not decompose the series, and can miss structure when long-term trends or seasonal components are present.

**CD-NOD** (Huang et al., 2020). A causal discovery method for heterogeneous and non-stationary data that combines invariance testing with constraint-based structure learning. CD-NOD does not model lagged

relationships explicitly; we evaluate it on a non-temporal version of the synthetic data to ensure a fair comparison.

**CFCI** (Peters et al., 2017). An adaptation of the Fast Causal Inference algorithm that accommodates latent confounders and selection bias, extended to time series via temporal tiering.

**CPC** (Colombo et al., 2012). A conservative variant of the PC algorithm optimized for temporal data, using temporal priority to restrict the search space. CPC outputs an extended CPDAG with ambiguous triples marked.

**BOSS-LiNGAM** (Shimizu et al., 2006). A two-step method combining CPDAG estimation via PC with ICA-based orientation under a linear acyclic non-Gaussian model.

**DYNOTEARS** (Pamfil et al., 2020). A score-based approach that estimates lagged and contemporaneous causal effects jointly by solving a penalized likelihood objective under a continuous acyclicity constraint.

Table 4: Hyperparameter settings. For synthetic datasets the lag is set to the true generative lag; for ETTh1 and the climate dataset the lag is fixed at 2.

| Method | Key parameters |
|---|---|
| **DCD (Ours)** | STL decomposition; CI test: partial correlation; significance threshold $p < 0.05$. |
| PCMCI+ | CI test: CMI-knn; $\alpha = 0.05$; majority collider rule; no FDR correction. |
| CD-NOD | CI test: partial correlation; adaptive thresholding; non-stationary surrogate tests. |
| CFCI | Conservative FCI variant; retains ambiguous colliders; respects temporal tiering. |
| CPC | Conservative PC; collider orientation only when unambiguous; outputs extended CPDAG; CI test: partial correlation; $\alpha = 0.05$. |
| BOSS-LiNGAM | Two-step: CPDAG estimation via PC, followed by ICA-based orientation; Anderson–Darling test ($\alpha = 0.05$); assumes linear acyclic models. |
| DYNOTEARS | Structural equation model with $\ell_1$ penalty $\lambda \in [0.01, 0.1]$; maximum lag $\tau \in \{2, 4, 6\}$; acyclicity tolerance and initialization per authors' defaults. |

All baselines are implemented using their official repositories with hyperparameters tuned following authors' recommendations.

## D Synthetic Data Generation

To evaluate the DCD framework with known ground truth, we generate synthetic multivariate time series that reflect autocorrelation, non-stationarity, and cross-variable dependence.

**Dataset configuration.** Each dataset consists of $n$ time points and $d$ variables, with sample sizes $n \in \{500, 1000, 1500\}$, dimensions $d \in \{4, 6, 8, 10\}$, and lag orders $l \in \{2, 3, 4\}$. Each variable combines trend, seasonality, and residual noise so that the resulting series exhibit both long-term dependencies and short-term causal interactions.

**Residual component.** The residual term captures short-term dependencies via an autoregressive process,

$$e_i(t) = \sum_{j=1}^{i-1} \sum_{l=0}^{3} \alpha_{ij}^l e_j(t-l) + \epsilon_i(t), \tag{24}$$

with $\alpha_{ij}^l \in [0.3, 0.8]$ and $\epsilon_i(t) \sim \mathcal{N}(0, 0.1)$. The summation produces instantaneous ($l = 0$) and lagged ($l = 1, 2, 3$) causal dependencies.

**Full series.** Each variable $X_i(t)$ is the sum of trend, seasonal, and residual components,

$$X_i(t) = \mu_i + \beta_i t + A_i \sin(2\pi t/\omega_i) + e_i(t), \tag{25}$$

with $\mu_i \in [5, 12]$, $\beta_i \in [0, 0.2]$, $A_i \in [0, 4]$, and $\omega_i \in [12, 25]$. This construction yields a controlled environment with known ground-truth causal relationships.

## E    Extended Ablation Results

This section provides the full tables and detailed analysis for the three ablation studies summarized in Section 4.6.

### E.1    Robustness to Decomposition Misalignment

Structural identifiability relies on consistent separation of components from the latent periodic driver $\Phi(t)$. We perturb the STL period $P$ around the true latent periodicities ($T \in \{15, 30\}$) across $d \in \{4, 6, 8\}$ (Table 5). Alignment with the fundamental frequency (**Optimal**, denoted as **Opt.** in Table 5) consistently minimizes spectral leakage and maximizes edge isolation. Conversely, we evaluate the framework under **Incorrect** (**Inc.**) specifications to test its resilience. At $d = 4$, an Optimal period $P = 30$ yields SHD $1.67 \pm 1.15$; even an Incorrectly specified $P = 10$ still gives SHD $8.33 \pm 1.15$, which remains lower than every baseline reported in Table 2. Accuracy therefore declines gracefully rather than catastrophically under misalignment: LOESS smoothing isolates the dominant non-stationary confounders even under sub-optimal bounds.

Table 5: Influence of the STL decomposition period $P$ on recovery across variable counts $d$. Results aggregated across sample sizes $n \in \{500, 1000, 1500\}$ at $\tau_{\max} = 4$.

| Variables ($d$) | Period ($P$) | Mean TPR ↑ | Mean FDR ↓ | Mean SHD ↓ |
|:---:|:---|:---:|:---:|:---:|
|  | 10 (Inc.) | $0.500 \pm 0.00$ | $0.757 \pm 0.04$ | $8.33 \pm 1.15$ |
|  | 15 (**Opt.**) | $0.750 \pm 0.00$ | $0.565 \pm 0.06$ | $5.00 \pm 1.00$ |
| **4** | 20 (Inc.) | $0.500 \pm 0.00$ | $0.747 \pm 0.03$ | $8.00 \pm 1.00$ |
|  | 30 (**Opt.**) | $0.750 \pm 0.00$ | $0.133 \pm 0.23$ | $1.67 \pm 1.15$ |
|  | 35 (Inc.) | $0.583 \pm 0.14$ | $0.720 \pm 0.06$ | $7.67 \pm 1.15$ |
|  | 10 (Inc.) | $0.722 \pm 0.10$ | $0.705 \pm 0.03$ | $12.00 \pm 1.00$ |
|  | 15 (**Opt.**) | $1.000 \pm 0.00$ | $0.578 \pm 0.04$ | $8.33 \pm 1.53$ |
| **6** | 20 (Inc.) | $0.722 \pm 0.10$ | $0.665 \pm 0.03$ | $10.33 \pm 1.15$ |
|  | 30 (**Opt.**) | $1.000 \pm 0.00$ | $0.490 \pm 0.09$ | $6.00 \pm 2.00$ |
|  | 35 (Inc.) | $0.667 \pm 0.17$ | $0.736 \pm 0.04$ | $13.00 \pm 1.00$ |
|  | 10 (Inc.) | $0.500 \pm 0.00$ | $0.608 \pm 0.06$ | $10.33 \pm 1.53$ |
|  | 15 (Inc.) | $0.458 \pm 0.07$ | $0.616 \pm 0.06$ | $10.33 \pm 1.53$ |
| **8** | 20 (**Opt.**) | $0.542 \pm 0.07$ | $0.468 \pm 0.07$ | $7.67 \pm 1.15$ |
|  | 30 (Inc.) | $0.417 \pm 0.07$ | $0.611 \pm 0.03$ | $10.00 \pm 1.00$ |
|  | 35 (Inc.) | $0.458 \pm 0.07$ | $0.662 \pm 0.08$ | $11.67 \pm 2.08$ |

*Note.* Alignment with the latent period optimizes accuracy; misaligned periods increase FDR through spectral leakage but avoid catastrophic loss of recall.

### E.2    Causal Depth and Dimensionality Scalability

We evaluate resilience to the FDR explosion typically observed in temporal models as the number of variables $d$ and the causal search space expand (Runge, 2020). We vary $d \in \{4, 6, 8\}$ and $\tau_{\max} \in \{2, 4, 6\}$, with results averaged over $n \in \{500, 1000, 1500\}$ (Table 6). SHD is essentially flat in the $d = 4$ and $d = 6$ regimes, and recall is maintained as $\tau_{\max}$ increases. Isolating short-term residuals thus prevents the propagation of spurious long-lag dependencies, even as dimensionality grows.

Table 6: Causal depth scalability: impact of search depth on DCD performance across variable counts $d$. Results aggregated across sample sizes $n \in \{500, 1000, 1500\}$ at optimal period $P = 30$.

| Variables ($d$) | Max lag ($\tau_{\max}$) | Mean TPR ↑ | Mean FDR ↓ | Mean SHD ↓ |
|:---:|:---:|:---:|:---:|:---:|
| **4** | 2 | $0.750 \pm 0.00$ | $0.217 \pm 0.20$ | $2.00 \pm 1.00$ |
| | 4 | $0.750 \pm 0.00$ | $0.133 \pm 0.23$ | $1.67 \pm 1.15$ |
| | 6 | $0.750 \pm 0.00$ | $0.300 \pm 0.09$ | $2.33 \pm 0.58$ |
| **6** | 2 | $1.000 \pm 0.00$ | $0.444 \pm 0.10$ | $5.00 \pm 1.73$ |
| | 4 | $1.000 \pm 0.00$ | $0.490 \pm 0.09$ | $6.00 \pm 2.00$ |
| | 6 | $1.000 \pm 0.00$ | $0.475 \pm 0.09$ | $5.67 \pm 2.08$ |
| **8** | 2 | $0.458 \pm 0.07$ | $0.576 \pm 0.02$ | $9.33 \pm 0.58$ |
| | 4 | $0.417 \pm 0.07$ | $0.611 \pm 0.03$ | $10.00 \pm 1.00$ |
| | 6 | $0.458 \pm 0.07$ | $0.630 \pm 0.06$ | $10.67 \pm 1.53$ |

*Note.* Stability of SHD under increasing $\tau_{\max}$ indicates that residual-level decomposition prunes spurious long-lag dependencies.

### E.3   Value of Multi-Scale Integration

To separate the contribution of STL preprocessing from multi-scale graph integration, we run PCMCI+ and DYNOTEARS on the STL residuals $R_i(t)$ directly, without the trend-proxy and seasonal-proxy edges added by DCD (Table 7). Decomposed inputs substantially improve both baselines (PCMCI+: TPR $0.45 \to 0.83$, SHD $28.5 \to 11.2$; DYNOTEARS: TPR $0.67 \to 0.79$, SHD $25.0 \to 14.8$), yet neither matches DCD in its full multi-scale configuration (TPR 1.00, SHD 6.0). The integration of time-proxy edges with residual-level structure is therefore a distinct source of DCD's gains, not an artifact of preprocessing.

Table 7: Isolation test: impact of multi-scale integration ($d = 6$, $n = 1000$; averaged over 3 seeds). "Raw" indicates inputs with trend and seasonality retained; "STL residuals" indicates inputs restricted to $R_i(t)$; "multi-scale" indicates the full DCD pipeline with time-proxy edges.

| Method | Input | TPR ↑ | FDR ↓ | SHD ↓ |
|:---|:---|:---:|:---:|:---:|
| PCMCI+ | Raw | $0.45 \pm 0.08$ | $0.81 \pm 0.02$ | $28.5 \pm 2.1$ |
| PCMCI+ | STL residuals | $0.83 \pm 0.11$ | $0.59 \pm 0.04$ | $11.2 \pm 1.6$ |
| DYNOTEARS | Raw | $0.67 \pm 0.05$ | $0.85 \pm 0.02$ | $25.0 \pm 1.8$ |
| DYNOTEARS | STL residuals | $0.79 \pm 0.06$ | $0.68 \pm 0.03$ | $14.8 \pm 1.7$ |
| **DCD (Ours)** | **Multi-scale** | $\mathbf{1.00 \pm 0.00}$ | $\mathbf{0.50 \pm 0.09}$ | $\mathbf{6.0 \pm 1.7}$ |

### E.4   Sensitivity to Spectral Separability

To test the graceful-decline claim from Section 4.6, we induce controlled spectral leakage $\varepsilon$ by introducing amplitude modulation into the seasonal component,

$$S_i(t) = A_i \big[1 + \lambda T_i(t)\big] \sin\left(\frac{2\pi t}{P}\right), \tag{26}$$

where $\lambda \in [0, 1]$ controls coupling strength. As $\lambda$ grows, the trend $T_i(t)$ modulates the seasonal amplitude, creating spectral sidebands that leak into the residual band and raising the measured $\varepsilon$. This form of violation models real-world phenomena such as climate oscillations with time-varying amplitude (e.g., El Niño Southern Oscillation intensity modulated by long-term ocean warming) or economic cycles whose magnitude changes with secular growth trends. The amplitude-modulated signal violates Assumption A2 (spectral separability) because energy originally concentrated at the fundamental frequency $2\pi/P$ now spreads into neighboring frequency bins, including those occupied by the residual estimator.

We conduct the experiment on synthetic data with $d = 6$ variables, $n = 1000$ time points, optimal period $P = 30$, and maximum lag $\tau_{\max} = 2$, averaging results over 3 random seeds. For each coupling level $\lambda$, we measure the realized cross-component correlation $\varepsilon = \max_{i,j} |\text{Cov}(\widehat{S}_i, \widehat{R}_j)|/(\sigma_{\widehat{S}_i} \sigma_{\widehat{R}_j})$ to quantify the actual leakage induced by the modulation.

Table 8 reports DCD performance across coupling levels. Even at the strongest coupling ($\lambda = 1.0$, measured $\varepsilon = 0.25$), DCD maintains TPR above 0.7, indicating that while leakage increases spurious discoveries, the mechanistic core $G_R^\star$ remains largely identifiable. This is consistent with Theorem 1, whose SHD bound scales linearly in $\varepsilon$ rather than discontinuously. The monotonic degradation (TPR $1.00 \to 0.72$, SHD $5.0 \to 13.5$) confirms that violations of A2 do not trigger catastrophic failure but rather proportional error growth, validating the framework's applicability to settings with moderate cross-frequency coupling.

Table 8: DCD performance under increasing spectral leakage (averaged over 3 seeds at $d = 6$, $n = 1000$, $P = 30$, $\tau_{\max} = 2$).

| Coupling $\lambda$ | Leakage $\varepsilon$ | Mean TPR ↑ | Mean FDR ↓ | Mean SHD ↓ |
|---|---|---|---|---|
| 0.0 (none) | 0.02 | $1.000 \pm 0.00$ | $0.444 \pm 0.10$ | $5.0 \pm 1.7$ |
| 0.2 (low) | 0.06 | $0.944 \pm 0.05$ | $0.512 \pm 0.06$ | $6.8 \pm 1.5$ |
| 0.5 (med) | 0.12 | $0.861 \pm 0.07$ | $0.589 \pm 0.05$ | $9.2 \pm 1.9$ |
| 1.0 (high) | 0.25 | $0.722 \pm 0.09$ | $0.694 \pm 0.04$ | $13.5 \pm 2.4$ |

### E.5 Sample Size Sensitivity

We evaluate the impact of sample size on causal discovery performance across $n \in \{500, 1000, 1500\}$ (Figure 4). This analysis addresses a fundamental question in time series causal discovery: does additional data consistently improve structure recovery, or do systematic biases dominate finite-sample effects?

**DCD performance and stability.** DCD exhibits exceptional stability across all sample sizes, maintaining TPR near 0.79 ($\pm 0.03$–0.05) with only marginal FDR growth from 0.36 to 0.40. The essentially flat TPR curve indicates that DCD's component-specific inference recovers the mechanistic core $G_R^\star$ even at $n = 500$, with additional samples providing marginal refinement rather than fundamental structural changes. This stability reflects the fact that once STL decomposition reliably isolates the residual component (which occurs at moderate $n$ for well-separated frequencies), the constraint-based search operates on a stationary weakly dependent process where the effective sample size grows linearly with $n$ (Lemma 2). The modest FDR increase ($0.36 \to 0.40$) likely reflects the fixed significance threshold $\alpha = 0.05$: as the effective degrees of freedom grow, borderline-significant edges cross the detection threshold, introducing a small number of false positives.

**Constraint-based baselines (PCMCI+, CFCI, CPC).** PCMCI+ shows unstable behavior across sample sizes: TPR fluctuates between 0.25 and 0.28, while FDR rises from 0.79 to 0.82 as $n$ increases from 500 to 1500. This counterintuitive trend—more data yielding higher false discovery rates—indicates that PCMCI+ is detecting shared autocorrelation and trend-driven covariation as causal structure. When applied to raw non-stationary data, the momentary conditional independence tests cannot distinguish between short-term mechanistic links and long-range correlation induced by common temporal drivers. CFCI and CPC exhibit even more severe degradation: both methods maintain FDR near 0.88–0.91 across all sample sizes, recovering fewer than 20–32% of true edges. The conservative orientation rules in these methods (retaining ambiguous colliders, restricting to extended CPDAGs) compound the difficulty of handling autocorrelation, producing sparse graphs that miss most lagged dependencies.

**BOSS-LiNGAM performance.** BOSS-LiNGAM recovers moderate TPR (0.31–0.34) but suffers from consistently high FDR (0.89–0.90). The two-stage procedure (CPDAG estimation via PC, followed by ICA-based orientation) is designed for cross-sectional data with independent observations. When applied to autocorrelated time series, the PC stage produces a dense skeleton because serial dependence between $X_i(t)$ and $X_j(t)$ at multiple lags is not screened off by conditioning sets drawn from the same time slice. The subsequent ICA orientation step cannot resolve these spurious edges, yielding high false discovery rates regardless of sample size.

**DYNOTEARS deterioration with sample size.** DYNOTEARS shows the most striking sample-size effect: TPR grows modestly from 0.44 to 0.50 as $n$ increases, but FDR remains fixed near 0.90–0.91 and SHD

grows dramatically from 28.4 to 35.2. This pattern is characteristic of overfitting in continuous-relaxation methods: the acyclicity penalty $h(\mathbf{W})$ is designed to encourage sparsity, but under strong autocorrelation the penalty becomes insufficient to suppress weakly supported edges. As $n$ grows, the optimization has more degrees of freedom to fit residual variance by introducing additional lagged connections, which increases both TPR (recovering more true edges) and SHD (adding many false edges). The net result is that DYNOTEARS benefits marginally from additional data in terms of recall, but the precision remains poor because the method cannot distinguish mechanistic causal links from autocorrelation-driven covariation.

**CD-NOD baseline behavior.** CD-NOD achieves the lowest SHD (9.67) across all sample sizes by construction: it discards all lagged edges and outputs only contemporaneous structure. The resulting graph is sparse and stable (TPR $\approx 0.23$, FDR $\approx 0.57$), but captures less than a quarter of the true causal relationships. This serves as a useful lower bound on SHD, confirming that methods achieving lower SHD than CD-NOD (such as DCD at $n = 500$: SHD 10.08) do so while recovering substantially more causal structure.

**Interpretation.** The flat or increasing FDR curves for PCMCI+, CFCI, CPC, BOSS-LiNGAM, and DYNOTEARS indicate that these methods are systematically biased when applied to non-stationary autocorrelated data: additional samples do not correct the bias because the bias arises from conflating different temporal scales. DCD's stable performance confirms that decomposition-based preprocessing removes the source of the bias, enabling consistent structure recovery across sample sizes.

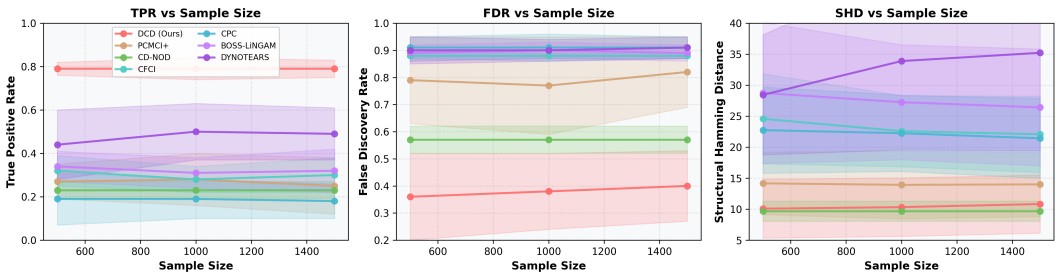

Figure 4: Impact of sample size on causal discovery performance. DCD achieves consistently high TPR and controlled FDR across $n \in \{500, 1000, 1500\}$, while baselines show unstable or deteriorating performance.

### E.6 Temporal Lag Effects

We evaluate the impact of increasing temporal lag $\tau_{\max} \in \{2, 3, 4\}$ on causal discovery performance (Table 1, Figure 5). The ground-truth generative model includes lagged dependencies up to $\tau = \tau_{\max}$, so methods must search over an expanding temporal window to recover the full causal structure. This analysis tests whether methods degrade gracefully as the search space grows, or suffer catastrophic failure due to the combinatorial explosion of candidate parent sets.

**DCD scalability.** DCD maintains TPR between 0.78 and 0.80 across all lag horizons, with FDR rising gradually from 0.28 to 0.48. The stable recall indicates that DCD successfully recovers mechanistic links at lags 2, 3, and 4 without suffering from the search-space explosion that affects constraint-based methods. The FDR increase reflects a fundamental tradeoff: as the maximum lag grows, the number of potential parents for each variable increases, and borderline edges near the significance threshold are more likely to pass the conditional independence tests due to random fluctuations. However, the growth is linear rather than exponential, and even at $\tau_{\max} = 4$ the FDR remains below 0.5, indicating that DCD recovers more true edges than false edges. The decomposition-based preprocessing is critical here: by isolating the residual component, DCD reduces the effective autocorrelation, which in turn reduces the number of spurious long-lag dependencies detected by the CI tests.

**PCMCI+ degradation.** PCMCI+ shows severe degradation at longer lags. TPR initially increases from 0.21 at lag 2 to 0.30 at lag 3, suggesting that the method begins to recover some true long-range dependencies. However, at lag 4, the TPR drops to 0.29 while FDR rises sharply to 0.86. This non-monotonic behavior

indicates that PCMCI+ is struggling to maintain conditional independence under expanding conditioning sets: as $\tau_{\max}$ grows, the number of lagged variables that must be conditioned upon increases, and the finite-sample reliability of the CMI-knn test decreases. The result is a proliferation of false positives (high FDR) alongside modest recall.

**Constraint-based baseline collapse.** CFCI and CPC show monotonically increasing SHD as lag grows (CFCI: $19.25 \rightarrow 26.00$; CPC: $17.08 \rightarrow 26.25$), with TPR remaining flat or declining slightly and FDR stable near 0.88–0.92. This pattern is characteristic of overfitting in constraint-based methods: the methods detect many edges (high SHD), but most are spurious (high FDR), and the true edges are missed (low TPR). The conservative orientation rules (CFCI retains ambiguous colliders; CPC outputs extended CPDAGs) prevent the methods from resolving the direction of edges, compounding the difficulty of recovering oriented lagged structure.

**BOSS-LiNGAM decline.** BOSS-LiNGAM exhibits steadily declining TPR ($0.35 \rightarrow 0.31$) and rising SHD ($21.58 \rightarrow 32.42$) as lag increases, with FDR stable near 0.89–0.90. The two-stage procedure (PC skeleton + ICA orientation) is fundamentally unsuited to lagged causal discovery: the PC stage does not respect temporal ordering, treating all lagged variables symmetrically, and the ICA orientation assumes acyclicity within each time slice, which is violated when lagged dependencies are present. The result is a dense skeleton with many false edges that cannot be oriented correctly.

**DYNOTEARS catastrophic failure.** DYNOTEARS shows the sharpest degradation of any method: TPR falls from 0.56 at lag 2 to 0.44 at lag 4, while FDR rises from 0.87 to 0.93 and SHD increases from 25.33 to 38.67. This behavior reflects the fundamental limitation of score-based continuous relaxations under temporal expansion: DYNOTEARS optimizes a separate weighted adjacency matrix $\mathbf{W}_\tau$ for each lag $\tau \in \{0, 1, \ldots, \tau_{\max}\}$, introducing $d^2 \cdot (\tau_{\max} + 1)$ free parameters. As $\tau_{\max}$ grows, the number of parameters increases linearly, and the acyclicity constraint $h(\mathbf{W}) = \mathrm{tr}(e^{\mathbf{W} \circ \mathbf{W}}) - d = 0$ becomes less effective at enforcing sparsity because the constraint operates on the aggregated graph $\mathbf{W} = \sum_\tau \mathbf{W}_\tau$ rather than on each individual lag matrix. The result is that DYNOTEARS admits many weakly supported edges to minimize residual variance, leading to catastrophic growth in false discoveries.

**CD-NOD lag-invariance.** CD-NOD remains completely flat across all lag values (TPR 0.23, FDR 0.57, SHD 9.67) because it does not model lagged relationships: the method outputs only contemporaneous structure, which is independent of $\tau_{\max}$. This serves as a useful control, confirming that the degradation observed in other methods is specifically due to their handling of temporal dependencies, not to other confounding factors.

**Interpretation.** The graceful degradation of DCD (linear FDR growth, stable TPR) contrasts sharply with the catastrophic failure modes observed in DYNOTEARS (exponential SHD growth) and the unstable behavior of PCMCI+ (non-monotonic TPR). This confirms that decomposition-based preprocessing is essential for robust causal discovery under expanding temporal horizons: by isolating the residual component, DCD suppresses the spurious long-lag autocorrelations that drive false discoveries in methods operating on raw data.

### E.7 Dimensionality Effects

We evaluate the impact of increasing the number of variables $d \in \{4, 6, 8, 10\}$ on causal discovery performance (Table 2, Figure 6). As dimensionality grows, the search space expands exponentially (each variable can have up to $d - 1$ parents at each of $\tau_{\max} + 1$ time lags), testing whether methods maintain accuracy or succumb to the curse of dimensionality.

**DCD robustness.** DCD maintains the best TPR/FDR balance across all dimensions: TPR remains between 0.76 and 0.84, with a modest decline as $d$ grows, while FDR rises from 0.20 at $d = 4$ to 0.44 at $d = 10$. The TPR decline is consistent with the expanding search space: as more variables are added, the ground-truth graph becomes sparser (the edge density $|E^\star|/d^2$ decreases), and some edges fall below the detection threshold due to finite-sample noise. However, DCD continues to recover more than three-

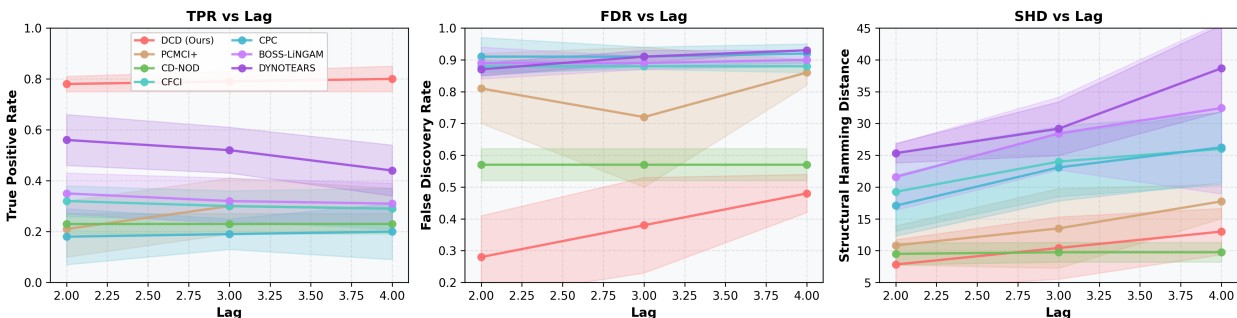

Figure 5: Effect of temporal lag on causal discovery performance. DCD maintains stable TPR and controlled FDR growth as lag increases from 2 to 4, while baselines show severe degradation (DYNOTEARS, PCMCI+) or flat performance due to discarding lagged structure (CD-NOD).

quarters of true edges even at $d = 10$, substantially outperforming all baselines. The FDR increase reflects the standard precision-recall tradeoff under high dimensionality: maintaining high recall requires accepting more borderline edges, some of which are false positives. Notably, DCD's FDR at $d = 10$ (0.44) remains far below the FDRs of PCMCI+ (0.79), CFCI (0.88), CPC (0.91), BOSS-LiNGAM (0.87), and DYNOTEARS (0.91), confirming that decomposition-based preprocessing provides a substantial advantage even in high-dimensional regimes.

**PCMCI+ dimensional instability.** PCMCI+ shows non-monotonic behavior as dimensionality increases: TPR drops sharply from 0.34 at $d = 4$ to 0.19 at $d = 6$, then recovers partially to 0.30 at $d = 10$. FDR remains high throughout (0.69–0.90), indicating that the method is detecting many spurious edges regardless of dimension. The non-monotonic TPR suggests that PCMCI+ is sensitive to the specific structure of the ground-truth graph at each dimension: at $d = 6$, the graph may contain particularly challenging conditional independence patterns (e.g., long conditioning sets, near-deterministic relationships) that cause the CMI-knn test to fail. At higher dimensions, the graph becomes sparser (fewer edges per variable), making some true edges easier to detect but also introducing more false positives due to random fluctuations in the test statistic.

**Constraint-based baseline failure.** CFCI and CPC show steadily deteriorating performance as dimensionality grows: SHD increases from $\approx 18$ at $d = 4$ to $\approx 30$ at $d = 10$ for CFCI, and from $\approx 18$ to $\approx 28$ for CPC. TPR remains low (0.20–0.31), and FDR remains high (0.88–0.95), indicating that these methods are recovering dense graphs with many false edges and missing most true edges. The conservative orientation rules (CFCI's collider retention, CPC's extended CPDAG output) are designed to avoid orientation errors, but they compound the difficulty of handling high-dimensional autocorrelated data: the methods output many unoriented edges (increasing SHD) while failing to resolve the true directed structure.

**BOSS-LiNGAM dimensional growth.** BOSS-LiNGAM shows increasing TPR as dimensionality grows ($0.26 \rightarrow 0.41$), but FDR remains consistently high (0.87–0.91) and SHD increases dramatically ($21.22 \rightarrow 37.00$). The increasing TPR suggests that BOSS-LiNGAM is recovering more true edges at higher dimensions, but the high FDR and SHD indicate that it is also adding many false edges. The two-stage procedure (PC skeleton + ICA orientation) scales poorly with dimension: the PC stage produces a dense skeleton due to autocorrelation, and the ICA orientation cannot prune the false edges because the non-Gaussianity assumption is violated by the residual components of the time series.

**DYNOTEARS dimensional collapse.** DYNOTEARS exhibits the most severe dimensional degradation: TPR drops from 0.75 at $d = 4$ to 0.22 at $d = 10$, while FDR rises from 0.81 to 0.91 and SHD increases from 13.67 to 35.00. At $d = 10$, DYNOTEARS is recovering fewer true edges than CD-NOD (which discards all lagged structure), while producing SHD more than three times higher. This collapse reflects the interaction between dimensionality and temporal lag: DYNOTEARS optimizes $d^2 \cdot (\tau_{\max} + 1)$ parameters (weighted adjacency matrices for each lag), which grows quadratically with $d$. At $d = 10$ and $\tau_{\max} = 4$, this

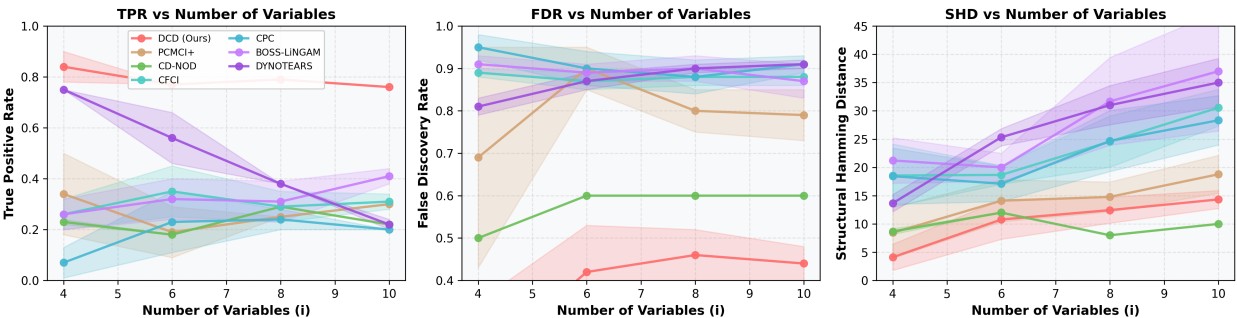

Figure 6: Effect of number of variables on causal discovery performance. DCD maintains the best TPR/FDR balance across all dimensions, while baselines show catastrophic degradation (DYNOTEARS) or high false discovery rates (PCMCI+, CFCI, CPC, BOSS-LiNGAM).

corresponds to 500 free parameters for a dataset with $n = 1000$ time points, giving an effective sample size of only 2 observations per parameter. The continuous acyclicity constraint cannot compensate for this severe undersampling, leading to catastrophic overfitting.

**CD-NOD dimensional behavior.** CD-NOD shows irregular behavior as dimensionality grows: TPR varies non-monotonically ($0.23 \rightarrow 0.18 \rightarrow 0.29 \rightarrow 0.22$), FDR remains constant (0.50–0.60), and SHD decreases from 8.67 at $d = 4$ to 8.00 at $d = 8$, then rises to 10.00 at $d = 10$. This pattern reflects the fact that CD-NOD outputs only contemporaneous structure: the non-monotonic TPR indicates that the method is more successful at recovering instantaneous links in some graph configurations than others, but the overall performance remains poor (recovering less than 30% of true edges).

**Interpretation.** The dimensional scaling analysis confirms that DCD's component-specific inference provides a fundamental advantage over methods operating on raw data. By isolating the residual component, DCD reduces the effective dimensionality of the search space: the trend and seasonal components are modeled as exogenous time-proxy edges rather than as lagged dependencies, which prevents the explosive growth in candidate parent sets that drives failure in DYNOTEARS and instability in PCMCI+. Even at $d = 10$, DCD maintains TPR above 0.75 and FDR below 0.45, substantially outperforming all baselines.

## F  Qualitative Analysis of Inferred Causal Graphs

Figure 7 compares the causal structures inferred by DCD and six baselines on the Arctic sea ice dataset. The comparison illustrates how the methods handle non-stationarity and autocorrelation.

**DCD (Figure 7a).** The recovered graph is sparse and physically interpretable. Modeling trend (T) and seasonal (S) components as distinct drivers isolates short-term mechanistic links in the residual layer. DCD recovers the documented SST–sea ice feedback oriented correctly in time and avoids the dense all-to-all connectivity produced when shared trends drive multiple variables. The multi-scale representation explicitly separates exogenous temporal forcing (time-proxy edges $t \rightarrow X$) from endogenous variable-to-variable interactions ($X \rightarrow Y$), enabling domain interpretation: the seasonal edges capture annual cycles in radiation, temperature, and precipitation; the trend edges capture long-term Arctic warming; and the residual edges capture month-to-month atmospheric and oceanic feedbacks.

**PCMCI+ (Figure 7b).** The graph contains ambiguous or unoriented edges (e.g. temperature–humidity) and misses the documented lagged SST $\rightarrow$ sea ice influence. When applied to raw non-stationary data, the strong seasonal signal masks subtler residual interactions: the momentary conditional independence tests detect the shared 12-month periodicity as a contemporaneous association rather than recognizing it as an exogenous driver. The result is a graph with many undirected or incorrectly oriented edges that conflate seasonal covariation with mechanistic causation.

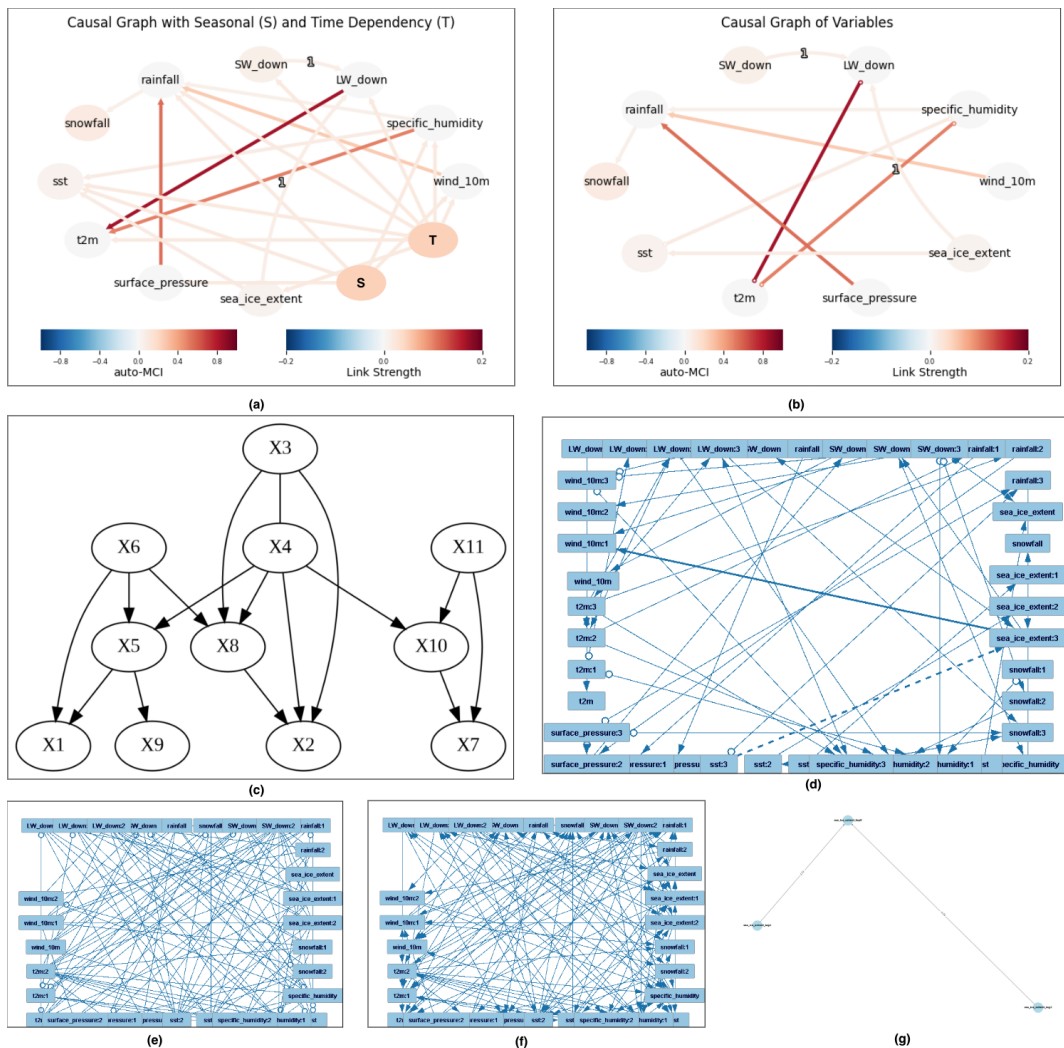

Figure 7: Causal graphs inferred by (a) DCD (ours), (b) PCMCI+, (c) CD-NOD, (d) BOSS-LiNGAM, (e) CFCI, (f) CPC and (g) DYNOTEARS. DCD captures seasonal (S) and time-dependent (T) influences explicitly.

**CD-NOD (Figure 7c).** CD-NOD identifies non-stationarity through invariance testing and correctly flags several variables as non-stationary, but it tends to link variables to the time index rather than to each other. Without explicit lagged modeling, the graph captures contemporaneous correlations (e.g. radiation $\leftrightarrow$ temperature) but misses time-delayed structure. The method outputs edges of the form $t \rightarrow X$ and $X \leftrightarrow Y$ (undirected due to potential latent confounding), which is conceptually similar to DCD's time-proxy edges but lacks the explicit multi-scale decomposition that separates trend, seasonal, and residual mechanisms.

**CFCI, CPC, BOSS-LiNGAM (Figures 7d–f).** These methods produce dense, overconnected graphs. The density is characteristic of false positives driven by autocorrelation: unfiltered serial correlation is misinterpreted as a causal link between variables with similar memory, giving graphs with limited domain interpretability. BOSS-LiNGAM in particular outputs a near-complete graph with edges between nearly all variable pairs, indicating that the PC skeleton stage detects many dependencies due to shared trends and seasonality, and the ICA orientation stage cannot resolve these spurious edges because the non-Gaussianity assumption is violated by the strongly periodic and non-stationary structure of the data. CFCI and CPC show similar overconnection, though with some edges left unoriented due to their conservative orientation rules.

**DYNOTEARS (Figure 7g).**  The recovered graph collapses to a trivial autoregressive structure: DYNOTEARS outputs only three nodes representing sea ice extent at lags 0, 1, and 2, with directed edges sea_ice_extent$_{t-1}$ → sea_ice_extent$_t$ (weight $-1.75$) and sea_ice_extent$_t$ → sea_ice_extent$_{t+2}$ (weight $-0.14$), while completely discarding all atmospheric and oceanic variables (surface pressure, wind velocity, humidity, temperature, radiation, precipitation, SST, salinity). This extreme sparsity indicates that the continuous acyclicity relaxation, when applied to strongly non-stationary and seasonal data, admits only the most dominant autocorrelation signal and fails to distinguish mechanistic climate interactions from persistence. The $\ell_1$ penalty $\lambda$ is designed to encourage sparsity, but under the combined influence of long-term warming trends (non-stationarity) and strong annual cycles (seasonality), the penalty becomes overly aggressive: it prunes not only spurious edges but also true mechanistic links, leaving only the univariate sea ice autoregression. This failure mode is consistent with DYNOTEARS' poor quantitative performance on the climate dataset (high FDR, low TPR in the main synthetic results) and confirms that score-based methods without explicit decomposition cannot reliably recover multi-scale causal structure in climate systems.

