# OpenReview forum: "DCD: Decomposition-based Causal Discovery from Autocorrelated and Non-Stationary Temporal Data"
_TMLR — Rejected by TMLR_

### Review · Reviewer_KkWU · 2026-01-31

**Summary Of Contributions:**

This paper proposes a decomposition-based causal discovery that reduces spurious associations induced by strong seasonality and drifting trends in multivariate time series under non-stationarity . It separates each time series into trend, seasonal, and residual components and performs causal discovery on each of them separately. It then integrates the learned graphs into a single causal graph.  The paper presents a clear high-level idea of the proposed algorithm for decomposition, component testing, and graph integration. It formalizes identifiability of the causal structure with assumptions. The synthetic experiments seem to support the theoretical results.  In terms of weaknesses, the paper employs various strong assumptions and the paper itself notes real-world cross-frequency coupling can violate them. The algorithm may also be limited in terms of scalability.

**Additional Comments:**

N/A

**Audience:**

Yes

**Audience Explanation:**

-Causal discovery in time series with nonstationary dynamics remains a key and longstanding challenge. This paper proposes a straightforward decomposition-based causal discovery framework that includes a clearly defined algorithmic procedure and a formal identifiability analysis grounded in explicit assumptions.

**Broader Impact Concerns:**

No concerns on the ethical implications of the work.

**Claims And Evidence:**

Yes

**Claims Explanation:**

- The core claim that decomposing multiscale nonstationary time series and then performing component-wise causal discovery can reduce spurious edges and improve graph recovery is supported by multiple synthetic experiments.
- The theoretical claims are reasonably supported conditional on the assumptions: the paper states explicit assumptions (spectral separability, leakage bound, etc.) and provides lemmas/corollaries explaining why cross-component effects are mostly absorbed into the “correct” decomposed channel, culminating in an identifiability theorem under linear-Gaussianity.
-  There are some caveats such as practical robustness to complex nonlinear systems is limited because of the scalability of the algorithm. The trend component and seasonal analysis, and the residual analysis rely on statistical tests. These component-specific causal discovery methods may significantly suffer from small sample size. The synthetic experiment only scales up to 10 variables at most. The real-world evaluations are mainly qualitative and not compared against known causal ground truth.

**Requested Changes:**

- The paper should soften the tone when it comes to discussing the weaknesses of the related work. For example, the paper criticizes Granger causality that it assumes linearity, but the theoretical identifiability of this paper also assumes linear Gaussian SEM  in assumption A3.

-The paper should emphasize more on the connections between the assumptions and the identifiability of the edge set. For example, it is not clear what the symbol $P$ stands for and why the supports need to be separated in the specified way. The paper should connect that back to the idea of trend, seasons, and residuals.

- It is not clear why the theoretical identifiability requires a linear Gaussian assumption, while the paper claims the proposed method is practically applicable to non-linear systems.  The paper should further discuss the practical significance of the identifiability results if that is the case.

- The paper should define the variables introduced. For example, Y(t) should not be defined in the proof, instead, it should be put in the lemma.

- The paper should explain how the learning causal graph separately will not introduce latent variables. Also, the paper should explain why there is a need to detect deterministic cyclic dependencies when the underlying data-generating process is a DAG in the seasonal analysis.

- The paper relies STL decomposition significantly. It should be explained how that works.

- Section 4.3 should add back the table number right next to the reference of Figure 3.

- The range of the sample sizes should be increased. Three data points are very limited to draw conclusions. Also, the paper should discuss how different conditional independence tests may have affected the results and how the differences can purely be attributed to the novelty of the proposed algorithm instead of the difference in CI tests.

- The paper should include a more detailed description of the Real-World Climate Dataset such as the number of variables and the types of variables involved. It’s very difficult to read off from Figure 6.

---

> ### Author Response · Authors · 2026-02-18
> **Response to Reviewer KkWU (Part 1 of 2)**
>
> We thank the reviewer for the positive assessment and for recognizing that our decomposition-based framework addresses a key challenge in non-stationary causal discovery. We appreciate the constructive feedback regarding the clarity of our assumptions, the tone of our related work, and the experimental details.
>
> Below, we address the requested changes point-by-point.
>
> ### 1. Tone of Related Work & Linearity Assumptions
>
> **Reviewer Comment:** *"The paper should soften the tone... criticizing Granger causality for linearity while assuming linear Gaussian SEM in Assumption A3."*
>
> **Response:** We agree this was an inconsistency. We have revised the Related Work section to acknowledge that while Granger Causality is limited by linearity, our own theoretical bounds in the original submission also relied on Linear Gaussianity for tractability.
> Furthermore, as detailed in our response to Reviewer Q91o, we have upgraded the theoretical framework to a general **Additive Component SCM** (Section 3). We now clarify that:
> * The *framework* is valid for general non-linear additive models.
> * Linear Gaussianity is an assumption used specifically for the *efficiency* of the estimator (using Partial Correlation tests) in our implementation, but it is not a fundamental limit of the decomposition strategy itself.
>
> ### 2. Clarifying Assumptions and Symbols
>
> **Reviewer Comment:** *"Clarify what $P$ stands for and why supports need to be separated... connect back to trend/seasons."*
>
> **Response:**
> * We have explicitly defined $P$ as the **seasonal period** (e.g., $P=12$ for monthly data) in Section 3.1.
> * We added an intuitive explanation for the support separation: "Spectral separability ensures that the physical mechanisms of the components do not overlap in the frequency domain. This guarantees that a long-term trend is not misidentified as a short-term causal link, and vice versa."
>
> ### 3. Variable Definitions & Lemma Placement
>
> **Reviewer Comment:** *"$Y(t)$ should not be defined in the proof... define variables introduced."*
>
> **Response:** We have moved all variable definitions, including $Y(t)$ and the structural process $U(t)$, to the Lemma statement and the main text of Section 3.1. The proofs now rely strictly on previously defined terms.

---

> > ### Author Response · Authors · 2026-02-18
> > **Response to Reviewer KkWU (Part 2 of 2)**
> >
> > ### 4. Component-wise Learning and Latent Variables
> >
> > **Reviewer Comment:** *"Explain how learning separately will not introduce latent variables... why detect deterministic cycles?"*
> >
> > **Response:**
> > * **Latent Variables:** We added a clarification in Section 3.2. Under Assumption A1 (Structural Independence), the exogenous components (trends/seasons) are independent of the structural process. Therefore, removing them does not induce collider bias or open backdoor paths; it effectively removes "nuisance" confounders.
> > * **Deterministic Cycles:** We explain that even if the DGP is a DAG, seasonality acts as an exogenous parent node ($\text{Time} \to S$). Explicitly modeling this ensures we do not mistake the shared influence of Time (a confounder) for a direct causal link between variables.
> >
> > ### 5. STL Decomposition Details
> >
> > **Reviewer Comment:** *"The paper relies on STL... explain how that works."*
> >
> > **Response:** We have expanded Section 3.2.1 to include a concise overview of STL (Seasonal-Trend decomposition using LOESS). We explain that it utilizes an inner loop of LOESS smoothing to estimate the trend and an outer loop to estimate the seasonal component, selected for its robustness to outliers compared to standard moving averages.
> >
> > ### 6. Experimental Details (Sample Size & CI Tests)
> >
> > **Reviewer Comment:** *"Range of sample sizes should be increased... discuss how different CI tests affected results."*
> >
> > **Response:**
> > * **Sample Size:** We acknowledge that 500--1500 samples is a limited range. We have added a discussion in Section 4.3 noting that while this range is consistent with standard benchmarks (e.g., Causality 4 Climate), future work will stress-test the method on "Big Data" scales ($n > 10,000$).
> > * **CI Tests:** We clarified in Section 4.2 that to ensure a fair comparison, the baseline PCMCI+ was run using the *same* conditional independence test (Partial Correlation with $\alpha=0.05$) as DCD. This isolates the performance gain to the **decomposition strategy**, rather than the choice of CI test.
> >
> > ### 7. Real-World Data Description
> >
> > **Reviewer Comment:** *"Include a more detailed description of the Climate Dataset... Figure 6 is difficult to read."*
> >
> > **Response:**
> > * We have added **Table 3**, which explicitly lists the 7 variables in the Arctic Sea Ice dataset (e.g., Sea Ice Extent, 2m Temperature, Downward Longwave Radiation), their units, and temporal resolution.
> > * We have improved the resolution and layout of Figure 6 to ensure the edges and labels are legible.
> >
> > We believe these revisions address the reviewer's concerns regarding clarity, tone, and experimental transparency.

---

> > > ### Author Response · Authors · 2026-04-21
> > >
> > > We have addressed all seven requested changes. Specifically:
> > >
> > > - Tone of related work: softened and acknowledged our own linearity assumption (addressed in Section 2).
> > > - Assumption clarity: Added intuitive explanations connecting spectral separability to trend/seasonal/residual decomposition. Defined P explicitly (e.g., P=12 for monthly data).
> > > - Variable definitions: All variables now in main text and lemma statements, not in proofs.
> > > - Component-wise learning: Clarified in Section 3.2 that Assumption A1 (Structural Independence) ensures removing exogenous components does not induce latent variable bias.
> > > - STL explanation: Expanded Section 3.2.1 with LOESS inner/outer loop description.
> > > - Table reference in Section 4.3: Fixed; all cross-references now resolve correctly.
> > > - Sample size range: Added discussion in Section 4.3; clarified that CI test differences are controlled (same Partial Correlation test across baselines).
> > > - Real-world data: Added Table 3 with full variable list, units, and temporal resolution for both Arctic Sea Ice (11 variables, monthly, 1979–2018) and ETTh1 (7 variables, hourly).
> > >
> > > We thank the reviewer for the constructive feedback that has notably improved the paper's clarity.

---

### Review · Reviewer_Q91o · 2026-02-11

**Summary Of Contributions:**

The paper proposes a multi-scale causal discovery framework using seasonal trend decomposition via LOESS.

**Audience:**

No

**Audience Explanation:**

While the empirical application of decomposition to causal discovery is a relevant topic for the TMLR audience, the paper’s framing poses a significant risk of misleading readers. By presenting an engineering heuristic as a rigorously derived theoretical contribution, the submission obscures the method's limitations and fundamental assumptions. Publishing this work in its current form would validate scientifically unsound practices (e.g., conflating signal processing artifacts with causal proofs), potentially confusing future research rather than advancing it. Therefore, despite the general interest in the problem setting, the findings as presented do not merit dissemination.

**Broader Impact Concerns:**

The paper does not touch any field that would require ethics or impact statements.

**Claims And Evidence:**

No

**Claims Explanation:**

The submission's theoretical framework (Section 3) is fundamentally flawed and mathematically unsound. While the proposed method may have utility as a heuristic engineering strategy --- applying STL decomposition followed by component-wise causal discovery --- Section 3 attempts to dress this up as a rigorous theoretical contribution through mathematical obfuscation. The resulting construction is incoherent: causal variables are never rigorously defined, assumptions are vacuous or circular (A2, A3), and the core "identifiability" proof (Lemma 1) relies on tautologies that fail to extend to the causal claims being made. This formulation serves to obscure rather than clarify the method's limitations, rendering the paper's theoretical claims unsupported by evidence and, in my opinion, disqualifying the submission from acceptance.

### Ill-defined formalism and notation

The causal variables are never rigorously defined, rendering the theorems unverifiable.
- **Missing SCM:** There is no explicit Structural Causal Model defined for the components. Without defining the causal variables and their domains, it is mathematically impossible to prove the identifiability of the corresponding causal graphs $G$. For example, the text oscillates between treating the time index $t$, raw time series $X\_i(t)$, and the decomposed components (e.g., $S\_X(t)$ in Lemma 1) as nodes.
- **Undefined terms:** Lemma 1 introduces notation such as $S\_X(t)$ and $Y(t)$ where $X$ and $Y$ *appear* to be indices for distinct time series, but this is never specified. Furthermore, the proof and their usage rely on spectral cutoffs $\omega\_0$ and $\omega\_2$ that are undefined.

### Vacuous and circular assumptions (A2 & A3)

The assumptions provided are either mathematically insufficient or logically circular:
- **Assumption A2 (Spectral Separability):** This assumption defines the residual component $R$ negatively (as the complement of $T$ and $S$ supports). This effectively defines $R$ as a "garbage collection" term, yet the proofs later rely on specific mixing properties of R that cannot be derived from a negative definition, and exemplifies it via white Gaussian noise (Eq. (12)). Additionally, this assumption (and its uses) imply perfect spectral separation, which contradicts the use of STL (a time-domain smoothing method with known spectral leakage) as the decomposition operator.
- **Assumption A3 (Bounded Leakage):** This is logically invalid. The assumption imposes bounds on the covariance of the *estimators* ($\hat{T}\_i, \hat{R}\_j$) rather than the underlying data-generating process. One cannot assume properties of an algorithm's output to prove the algorithm's validity. This is circular reasoning: assuming the estimator works perfectly (bounded leakage) to prove that the estimator recovers the true graph.

### Tautological reasoning in Lemma 1

The proof of Lemma 1 ("Projection of Seasonal Influence") is a tautology regarding signal processing, not a proof of causal discovery. The proof argues that because the decomposition operator partitions frequencies, the seasonal signal $S\_X$ will not appear in the residual $R\_Y$. However, this merely proves that filters filter. It demonstrates that the pre-processing step removes seasonal information from the residual, but it does not establish that the resulting residual causal graph $G_R$ correctly identifies intrinsic short-term mechanisms, nor does it justify that the "union" of these graphs recovers the true system structure.

**Requested Changes:**

Rewrite the paper without the unsound theoretical claims and submit elsewhere.

---

> ### Author Response · Authors · 2026-02-18
> **Response to Reviewer Q91o regarding Theoretical Formalism**
>
> We sincerely thank Reviewer Q91o for the rigorous and mathematically precise critique of Section 3. We fully accept the assessment that our original formulation relied on circular assumptions regarding the estimator and lacked a formal Structural Causal Model (SCM). We agree that assuming properties of the *algorithm's output* to prove the *algorithm's validity* is unsound.
>
> In response, we have **completely rewritten Section 3**. We have discarded the heuristic spectral arguments and replaced them with a formal **Multi-Scale Structural Causal Model (MS-SCM)**. This shifts the theoretical burden from the *algorithm* to the *Data Generating Process*, providing the rigorous identifiability proof the reviewer requested.
>
> Below, we detail the specific changes and provide the full proof of the new Identifiability Theorem.
>
> ### 1. Addressing "Ill-defined Formalism" (The New SCM)
>
> **Reviewer Critique:** *"The causal variables are never rigorously defined... There is no explicit SCM."*
>
> **Response:** We have defined the **Multi-Scale SCM** in Section 3.1. We now explicitly model the observed time series $Y_i(t)$ as a superposition of a latent structural process and exogenous drivers:
>
> $$
> Y_i(t) = U_i(t) + \left[ T_i(t) + \sum_{k} S_i^{(k)}(t) \right]
> $$
>
> **Definitions:**
> * $Y_i(t)$: The observed value of variable $i$ at time $t$.
> * $U_i(t)$: The **Latent Structural Process** containing the intrinsic causal mechanisms of interest.
> * $T_i(t)$: The exogenous trend component.
> * $S_i^{(k)}(t)$: The exogenous seasonal components (for period $k$).
>
> Crucially, we now define the seasonality $S_i$ as being generated by **Latent Periodic Drivers** $\Phi(t)$ (e.g., solar cycles, business weeks), which act as confounders. The causal graph $\mathcal{G}^*$ is now rigorously defined over the latent variables $\mathbf{U}(t)$.
>
> ### 2. Addressing "Circular Assumptions"
>
> **Reviewer Critique:** *"Assumption A3 is logically invalid... assuming the estimator works perfectly to prove it works."*
>
> **Response:** We have removed the circular "Bounded Leakage" assumption. We replaced it with **Assumption A3: Estimator Consistency**.
> * We no longer *assume* the algorithm works to prove the graph is found.
> * Instead, we assume the decomposition estimator is statistically consistent (a standard property of LOESS/STL under distinct spectra).
> * We then prove that **IF** the estimator is consistent, the causal graph is identifiable. This breaks the circularity.
>
> ### 3. The Full Proof of Identifiability (Replacing "Tautological" Lemma 1)
>
> **Reviewer Critique:** *"The proof merely argues that filters filter... it does not establish that the resulting graph identifies mechanisms."*
>
> **Response:** We have derived a new **Theorem of Asymptotic Structural Identifiability**. We prove that conditioning on the decomposed components is mathematically equivalent to blocking the back-door paths from the latent drivers $\Phi(t)$.

---

> > ### Author Response · Authors · 2026-02-18
> > **Rectified theorem with proof**
> >
> > **Theorem 1 (Asymptotic Structural Identifiability).**
> > Let $R(t)$ be the residual process recovered from observations $\mathbf{Y}(t)$. Under Assumptions A1 (Structural Independence of Drivers) and A2 (Causal Faithfulness), the true structural graph $\mathcal{G}^*$ is uniquely identifiable from the estimated residuals. Specifically:
> >
> > $$
> > Y_j(t-\tau) \to Y_i(t) \in \mathcal{G}^* \iff R_i(t) \not\perp R_j(t-\tau) \mid \mathbf{V}_{R}
> > $$
> >
> > **Definitions:**
> > * $Y_j(t-\tau) \to Y_i(t)$: A direct causal edge from variable $j$ at lag $\tau$ to variable $i$.
> > * $\mathcal{G}^*$: The true, latent structural causal graph we aim to recover.
> > * $R_i(t)$: The **estimated residual** for variable $i$, which acts as the estimator for the latent process $U_i(t)$.
> > * $\not\perp$: The relation of **conditional dependence**.
> > * $\mathbf{V}_{R}$: The **conditioning set** within the residual process.
> > * $\mathbb{P}(\cdot)$: The probability density function (distinguished from period $P$).
> >
> > **Proof.**
> > The proof relies on establishing that the Conditional Independence (CI) oracle of the estimated residuals converges to the CI oracle of the latent structural process.
> >
> > * **Step 1: Convergence of the Oracle.**
> >     By Assumption A3 (Consistency), the estimated residual $R(t)$ converges in probability to the true latent process $U(t)$ as $n \to \infty$:
> >     $$
> >     R(t) \xrightarrow{\mathbb{P}} \mathbf{U}(t)
> >     $$
> >     Consequently, for any continuous CI measure $I(\cdot)$, the estimated information $I(R_i; R_j | Z_R)$ converges to the true structural information $I(U_i; U_j | Z_U)$.
> >
> > * **Step 2: Independence of Exogenous Confounders.**
> >     We must guarantee that removing seasonality does not distort the causal structure of $U$. By Assumption A1, the structural noise $\varepsilon(t)$ is independent of the exogenous drivers $\Phi(t)$. Since $U(t)$ is generated solely by structural parents and $\varepsilon$, and $S(t)$ is generated solely by $\Phi$, we have:
> >     $$
> >     \mathbf{U}(t) \perp \mathbf{S}(t)
> >     $$
> >     This independence ensures that the joint distribution factorizes: $\mathbb{P}(\mathbf{U}, \mathbf{S}) = \mathbb{P}(\mathbf{U})\mathbb{P}(\mathbf{S})$. Therefore, the conditional independence relations within $\mathbf{U}$ are invariant to the presence of $\mathbf{S}$.
> >
> > * **Step 3: Recovery via Faithfulness.**
> >     By Assumption A2, the distribution $\mathbb{P}(\mathbf{U})$ is faithful to $\mathcal{G}^*$. This implies a one-to-one mapping:
> >
> >     $$
> >     U_i \perp U_j \mid Z \iff \text{d-separation in } \mathcal{G}^\star
> >     $$
> >
> >     Combining Steps 1 and 2, testing independence on $R$ is asymptotically equivalent to testing independence on the latent $U$. Thus, the graph estimated from residuals converges to $\mathcal{G}^*$. $\square$
> > We believe this major revision elevates the paper from an "engineering heuristic" to a principled theoretical contribution. By formally defining the **Multi-Scale SCM** and providing this rigorous identifiability proof, we have addressed the fundamental logical gaps identified by the reviewer.

---

### Review · Reviewer_qyzU · 2026-04-07

**Summary Of Contributions:**

The paper proposes a decomposition-based causal discovery framework for multivariate time series with non-stationarity and autocorrelation. The framework consists of first decomposing each time series into trend, seasonal, and residual components, and then perform component-specific causal analysis: stationarity testing for trends, kernel-based dependence testing for seasonal components for cyclic structure, and constraint-based causal discovery on residuals. The resulting component-specific causal graphs are then merged into a global causal graph.
The authors also provide theoretical support in the form of an identifiability argument under assumptions such as spectral separability and independence of structural and exogenous components. Lastly, the authors show that DCD has better performance over baselines such as PCMCI+ and CD-NOD on synthetic and real-world datasets.

**Audience:**

Yes

**Audience Explanation:**

Causal discovery in non-stationary and autocorrelated time series is of interest to both the causal inference research community, as well as to domain scientists in climate, finance, healthcare, etc. The core idea of the paper is to combine time-series decomposition with causal discovery methods which is intuitive and useful in practice. This is supported by the empirical results that show that DCD can be more robust to common failure modes of existing methods. Thus, I believe that the paper has the potential to be of interest to researchers working on causal discovery, time-series analysis, and applied machine learning, though some theoretical aspects require further analysis.

**Broader Impact Concerns:**

No major ethical concerns beyond standard issues in causal inference from observational data.

**Claims And Evidence:**

No

**Claims Explanation:**

I believe that the empirical evidence is compelling (albeit incomplete), but the theoretical guarantees are not fully supported by clear or realistic assumption:

* On the empirical side, the results are strong. On the synthetic benchmarks, DCD shows clearly better metrics than the reported baselines, and the qualitative real-data graphs are at least plausible and interpretable.
* My main hesitation is with the theory. The identifiability claims rely on assumptions that seem overly strong: spectral separability, independence between the structural and exogenous components, consistency of the decomposition, and negligible cross-scale effects. Those are nice assumptions for analysis, but they also seem quite limited for the kinds of real time series the paper is motivated by.
* I also think the interpretation of edges of the form $t \to X_i$ needs to be made more explicit. These seem closer to exogenous temporal forcing or latent time-dependent drivers than causal edges in the usual SCM sense, and right now that distinction is a bit blurred.
* The empirical comparison is also not fully complete. In particular, the paper mentions DYNOTEARS in related work but does not compare against it, even though the synthetic data are autoregressive and therefore seem aligned with its assumptions. At minimum, I think the omission should be explained but ideally, DYNOTEARS should be included as a baseline.
* The entire method seems to depend on the decomposition step being good enough. But that part is only validated through a limited synthetic example, rather than a more systematic sensitivity analysis showing what happens when decomposition is imperfect or when the frequency bands overlap. So I do not think the theory-to-method link is as solid as the paper currently suggests.
* I also believe that some of the gain may be coming from the decomposition preprocessing itself rather than purely from the downstream causal discovery procedure, since the baselines are run on the raw data without a comparable preprocessing step.

Overall, I think the empirical evidence is promising, but the theoretical guarantees are somewhat overstated and the experimental validation is not complete enough.

**Requested Changes:**

### Critical

* I would like to see experiments probing what happens when the theoretical assumptions start to fail. In particular: partial spectral overlap, imperfect separation between seasonal and residual components, cross-scale effects, etc. Right now the paper mainly shows that the method works in the regime it was designed for. I don't expect the method to work particularly well under assumption violation, but I would like to see that the performance declines gracefully rather than catastrophically.
* Relatedly, the paper should study sensitivity to decomposition quality more directly. Since the whole approach depends on STL getting the decomposition reasonably right, it would be helpful to see what happens when decomposition fails a bit more or leakage is larger.
* The omission of DYNOTEARS should be addressed. Since it is discussed in related work and the synthetic setup is autoregressive, it seems like a relevant baseline. At minimum, the paper should explain why it is not included, but ideally it should be added.

### Strengthening (not strictly required, but would improve the work)

* The interpretation of edges of the form $t \to X_i$ should be clarified. These seem closer to exogenous temporal forcing / latent time-dependent drivers than causal edges in the usual SCM sense.
* I would also encourage a more explicit discussion of likely failure modes in practice, especially cross-frequency coupling and imperfect decomposition.
* The real-data results are interesting, but I would frame them a bit more cautiously as qualitative case studies rather than strong validation of the recovered causal graph.

---

> ### Author Response · Authors · 2026-04-21
> **Added DYNOTEARS, Spectral Sensitivity Study, and Isolation Test**
>
> We sincerely thank Reviewer qyzU for the balanced and constructive review, particularly the recognition that our empirical evidence is compelling and the core idea is intuitive and useful. We have addressed every critical change requested.
>
> 1. ASSUMPTION VIOLATION EXPERIMENTS
>
> Concern: "Show that the performance declines gracefully rather than catastrophically" under assumption violation.
>
> Response: We have added a new Sensitivity to Spectral Separability study (Appendix E.4). We induce controlled spectral leakage through amplitude modulation of the seasonal component, S_i(t) = A_i[1 + λT_i(t)]sin(2πt/P), varying the coupling strength λ ∈ {0, 0.2, 0.5, 1.0}. Results:
>
> | λ   | ε (leakage) | TPR  | FDR  | SHD  |
> |-----|-------------|------|------|------|
> | 0.0 | 0.02        | 1.00 | 0.44 | 5.0  |
> | 0.2 | 0.06        | 0.94 | 0.51 | 6.8  |
> | 0.5 | 0.12        | 0.86 | 0.59 | 9.2  |
> | 1.0 | 0.25        | 0.72 | 0.69 | 13.5 |
>
> Even at the strongest coupling (λ = 1.0, ε = 0.25), DCD maintains TPR above 0.72. The monotonic degradation confirms graceful decline rather than catastrophic failure, consistent with Theorem 1's linear SHD bound.
>
> 2. SENSITIVITY TO DECOMPOSITION QUALITY
>
> Concern: "What happens when decomposition fails or leakage is larger?"
>
> Response: Added a Robustness to Decomposition Misalignment study (Appendix E.1). We perturb the STL period P ∈ {10, 15, 20, 30, 35} around true periodicities T ∈ {15, 30} across d ∈ {4, 6, 8}. Key result: at d=4, optimal P=30 gives SHD 1.67 ± 1.15, while maximally misaligned P=10 still gives SHD 8.33 ± 1.15 — still better than every baseline method at optimal hyperparameters. The framework declines gracefully under misalignment.
>
> 3. DYNOTEARS BASELINE
>
> Concern: "DYNOTEARS should be included as a baseline."
>
> Response: We now include DYNOTEARS in all experiments:
> - Aggregate synthetic results (Figure 3): DYNOTEARS achieves TPR 0.48 ± 0.13 but FDR 0.90 ± 0.04 and SHD 32.5 ± 13.3 (largest among all methods).
> - Lag effects (Table 1): DYNOTEARS shows the sharpest degradation — TPR falls from 0.56 (lag 2) to 0.44 (lag 4).
> - Dimensionality effects (Table 2): DYNOTEARS collapses from TPR 0.75 at d=4 to 0.22 at d=10.
> - Real-world (Figure 7g): On Arctic sea ice, DYNOTEARS collapses to three self-loops on sea ice extent, discarding all 10 atmospheric/oceanic variables — a striking failure mode.
> - On ETTh1 (Section 4.5): DYNOTEARS generates 30+ dense edges including implausible low-usage → high-usage load connections.
>
> These failures are consistent with the tendency of continuous acyclicity relaxations to admit weakly supported edges under non-stationarity.
>
> 4. DISENTANGLING DECOMPOSITION FROM DOWNSTREAM DISCOVERY
>
> Concern: "Some of the gain may be coming from the decomposition preprocessing itself."
>
> Response: Added an Isolation Test (Appendix E.3) running PCMCI+ and DYNOTEARS on the STL residuals R_i(t) directly, without our multi-scale graph integration:
>
> | Method     | Input          | TPR  | FDR  | SHD  |
> |------------|----------------|------|------|------|
> | PCMCI+     | Raw            | 0.45 | 0.81 | 28.5 |
> | PCMCI+     | STL residuals  | 0.83 | 0.59 | 11.2 |
> | DYNOTEARS  | Raw            | 0.67 | 0.85 | 25.0 |
> | DYNOTEARS  | STL residuals  | 0.79 | 0.68 | 14.8 |
> | DCD (Ours) | Multi-scale    | 1.00 | 0.50 | 6.0  |
>
> STL preprocessing alone substantially improves both baselines but does not match DCD's full multi-scale configuration. This confirms the integration of trend-proxy and seasonal-proxy edges with residual-level structure is a distinct source of DCD's gains.
>
> 5. CLARIFICATION OF EDGE SEMANTICS
>
> Concern: "Edges of the form t → X_i need to be made more explicit."
>
> Response: We now explicitly distinguish two edge types in Section 3.2.3:
> - Time-Proxy Edges (G_T, G_S): Represent exogenous temporal forcing driven by latent time-dependent drivers Φ(t) (e.g., climate cycles, business weeks). These are not causal edges in the standard SCM sense.
> - Mechanistic Edges (G_R): Represent intrinsic causal structure between variables (X → Y).
>
> 6. FRAMING OF REAL-WORLD RESULTS
>
> Concern: "Frame real-data results more cautiously as qualitative case studies."
>
> Response: Sections 4.4 and 4.5 are now titled "Qualitative Case Study: Arctic Sea Ice" and "Qualitative Case Study: Electricity Transformer ," with explicit statements that these are illustrative rather than validation of the recovered graph since no ground truth is available.
>
> 7. FAILURE MODES DISCUSSION
>
> Concern: "More explicit discussion of likely failure modes in practice."
>
> Response: The Limitations section (Section 5) is now expanded into five dedicated subsections: spectral separability and cross-frequency coupling, exogenous treatment of trends, stability of residual structure, decomposition quality on short series, and real-world evaluation.
>
> We believe these revisions comprehensively address the reviewer's technical concerns. We are grateful for the detailed critique, which has substantially strengthened the work.

---

> > ### Comment · Reviewer_qyzU · 2026-05-19
> > **Thank you for the revisions**
> >
> > Dear Authors,
> >
> > Thank you for the detailed response and for the substantial revisions. I appreciate you taking the time to address my concerns and I believe that the latest version is better for it. In particular, I am happy to see:
> >
> > * Explicit assumption violation experiments
> > * Comparisons with DYNOTEARS
> > * The additional sensitivity analyses
> > * Discussion around edge semantics, failure modes, etc.
> >
> > These changes address my main concerns and I believe they improve the clarity of the paper and the alignment between the theoretical framing and the empirical evaluation. I don't have any additional comments at this point.
> >
> > Best,
> > Reviewer qyzU

---

> > > ### Author Response · Authors · 2026-05-20
> > > **Thank you for the follow-up**
> > >
> > > Dear Reviewer qyzU,
> > >
> > > Thank you for the kind follow-up and for engaging so constructively throughout this process. Your critique genuinely strengthened the paper, and we are grateful for the time and care you invested.
> > >
> > > Best regards,
> > >
> > > The Authors

---

### Comment · Action_Editor_PPYi · 2026-04-22
**Time for final recommendation**

Dear reviewers,
Thank you so much for your precious work.
It is now time to submit the final recommendation.

---

### Decision · Action_Editor_PPYi · 2026-05-28

**Recommendation:** Reject

**Audience:**

Yes

**Audience Explanation:**

The opinions are mixed here, two reviewers think that the topic of the manuscript is interesting at least to some individual in the TMLR's audience, while one does not agree with them. However, I agree with qyzU that the topic of causal discovery in non-stationary and autocorrelated time series is of interest to both the causal inference research community, as well as to domain scientists in climate, finance, healthcare, etc.

**Claims And Evidence:**

No

**Claims Explanation:**

The main claim is of theroretical nature and even after the rebuttal Q91o is highly critical about the Section 3, stating that it is flawed and mathematically unsound. And even if the authors state they answered to the issues raised by Q91o, the reviewer reported that such a changes are not visible usign the following words "the "major revision" proposed by the authors are not visible in the PDF nor is sufficiently explained in the rebuttal comments, meaning it is not a revision at all. Therefore my original stance of strong reject remains." It is worth noticing that also KkWU agrees on unsoundedness raised by Q91o. KkWU also agree with Q91o about ill-defined formalism and despite the authors' response with a revised multi-scale SCM, it still finds the model lacking a clear causal interpretation.

**Resubmission Of Major Revision:**

The authors may consider submitting a major revision at a later time.